

Jesús Vergara Temprado

# Contribution of feldspar and marine organic aerosols to global ice nucleating particle concentrations

Jesús Vergara-Temprado[1], Theodore W. Wilson[1], Daniel O'Sullivan[1], Jo Browse[1,2], Kirsty J. Pringle[1], Karin Ardon-Dryer[3], Allan K. Bertram[4], Susannah M. Burrows[5], Darius Ceburnis[6], Paul J. DeMott[7], Ryan H. Mason[4], Colin D. O'Dowd[6], Matteo Rinaldi[8], Benjamin J. Murray[1], and Ken S. Carslaw[1]

[1]Institute for Climate and Atmospheric Science, School of Earth and Environment, University of Leeds, Woodhouse Lane, Leeds, LS2 9JT, UK
[2]College of Life and Environmental Sciences, University of Exeter, Penryn, TR10 9EZ, UK
[3]Department of System Biology, Harvard University, Harvard Medical School, Boston, USA
[4]Department of Chemistry, University of British Columbia, Vancouver, BC, V6T1Z1, Canada
[5]Pacific Northwest National Laboratory, Atmospheric Sciences and Global Change Division, P.O. Box 999 MS K-24, Richland, WA 99352, USA
[6]School of Physics and Centre for Climate and Air Pollution Studies, Ryan Institute, National University of Ireland Galway, Galway, Ireland.
[7]Department of Atmospheric Science, Colorado State University, Fort Collins, CO 80523-1371
[8]Italian National Research Council (CNR) - Institute of Environmental Sciences and Climate (ISAC), via P. Gobetti 101, 40129 Bologna, Italy

*Correspondence to:* Jesús Vergara Temprado (eejvt@leeds.ac.uk)

**Abstract.** Ice nucleating particles (INP) are known to affect the amount of ice in mixed-phase clouds, thereby influencing many of their properties. The atmospheric INP concentration changes by orders of magnitude from terrestrial to marine environments, which typically contain much lower concentrations . Many modelling studies use parameterizations for heterogeneous ice

nucleation and cloud ice processes that do not account for this difference because they were developed based on measurements predominantly from terrestrial environments. Errors in the assumed INP concentration will influence the simulated amount of ice in mixed-phase clouds, leading to errors in top-of-atmosphere radiative flux and ultimately the climate sensitivity of climate models. Here we develop a global model of INP concentrations relevant for mixed-phase clouds based on laboratory and field measurements of ice nucleation by K-feldspar (an ice-active component of desert dust) and marine organic aerosols (from sea

spray). The simulated global distribution of INP concentrations based on these two-species agrees much better with currently available ambient measurements than when INP concentrations are assumed to depend only on temperature or particle size. Underestimation of INP concentrations in some terrestrial locations may be due to neglect of INP from other terrestrial sources. Our model indicates that, on a monthly or yearly average basis, desert dusts dominate the contribution to the INP population over much of the world, but marine organics become increasingly important in the world's remote oceans and can dominate in

the Southern Ocean at some time of the year. Furthermore, we show that day-to-day variability is important and since desert dust aerosol tends to be sporadic, marine organics dominate the INP population on many days per month in much of the mid and high latitude northern hemisphere. This study advances our understanding of which aerosol species need to be included in



order to adequately describe the global and regional distribution of INP in models, which will guide ice nucleation researchers on where to focus future laboratory and field work.

## 1 Introduction

In the absence of aerosol particles which can act as ice nucleating particles (INP), liquid water droplets can supercool to temperatures below $-37^{o}C$ (Riechers et al., 2013; Herbert et al., 2015). It is well-known that ice formation frequently occurs at much higher temperatures in many clouds around the globe, indicating that INP are present to a greater or lesser extent depending on the location and the aerosol properties (Choi et al., 2010; Rosenfeld et al., 2011). In supercooled and mixed-phase clouds (containing ice and water) INP cause clouds to glaciate, which leads to changes in many cloud properties such as cloud lifetime, their radiative effect on the atmosphere, and the formation of precipitation through the Wegener–Bergeron–Findeisen process (Murphy and Koop, 2005; Korolev, 2007) and possibly other cloud ice multiplication processes (Hallett and Mossop, 1974). In the mixed-phase cloud regime the dominant freezing mechanism is thought to be through INP that are immersed within cloud droplets, known as immersion freezing (Westbrook and Illingworth, 2011; Field et al., 2012; Murray et al., 2012). Hence, this is the pathway we focus on in this study.

The current representation of heterogeneous freezing in climate models and operational numerical weather prediction models is usually based on parameterizations that depend on the temperature (Young, 1974; Meyers et al., 1992) or the size distribution of aerosol particles as well as the temperature (DeMott et al., 2010). However, these parameterizations treat aerosol all around the globe and across seasons as having the same ice nucleating properties. However, studies have shown that clouds are sensitive to INP concentrations, which could affect the radiative balance of the atmosphere (Zeng et al., 2009; Hoose et al., 2010b; DeMott et al., 2010; Wang et al., 2014; Tan et al., 2016). Over the Southern Ocean clouds tend to persist in a supercooled state more commonly than models predict (Bodas-Salcedo et al., 2014), which might be related to very low INP concentrations in this region. The variability among different models in the representation of cloud glaciation can lead to differences of 10s of degrees in the temperature at which clouds glaciate (McCoy et al., 2015b). A poor representation of mixed-phase clouds in climate models has been shown to be important for climate prediction. For example, Tan et al. (2016) concluded that the response of global mean surface temperature to a doubling of $CO_2$ is more than one degree greater when mixed-phase clouds are better represented. This cloud-phase feedback is particularly sensitive to the amount of supercooled liquid in Southern Ocean mixed-phase clouds where most current models are biased relative to observations (McCoy et al., 2015b).

In future, regional and global climate models will include improved representations of cloud processes (Bauer et al., 2015), including ice processes, so an improved representation of heterogeneous ice nucleation will be required to make the models more physically realistic and correct some of the main model biases. In particular, studies have shown that clouds are sensitive to INP concentrations, which could affect the radiative balance of the atmosphere (Zeng et al., 2009; Hoose et al., 2010b; DeMott et al., 2010; Wang et al., 2014; Tan et al., 2016). The reliability of such studies will depend on being able to relate the changes in cloud properties to emitted aerosol species so that we can attribute future changes in weather and climate to particular aerosol sources. A similar approach has been used in global aerosol models for many years, enabling aerosol radiative





forcing to be related to anthropogenic and natural emissions and their effects on aerosols and cloud droplet formation (Ghan and Schwartz, 2007; Rap et al., 2013; Carslaw et al., 2013; Kodros et al., 2015). Our ability to achieve the same level of realism for ice formation has been much more difficult to achieve, partly because it has been challenging to identify species-specific ice nucleating properties (Hoose and Möhler, 2012; Murray et al., 2012) and model them on a global scale.

Previous studies have simulated heterogeneous ice nucleation on a global scale accounting for different aerosol species (Hoose et al., 2010b, a; Spracklen and Heald, 2014). These studies used classical nucleation theory to calculate nucleation rates using contact angles derived from laboratory data for each INP species. This approach has the advantage that the time dependence of ice nucleation is represented, but when a single contact angle is used to describe ice nucleation by a single aerosol species, particle-to-particle variability is not represented (Herbert et al., 2014). Classical nucleation theory can be

extended with a distribution of contact angles to account for differences in the ice nucleating ability between different particles within the same material (Niedermeier et al., 2011; Broadley et al., 2012; Herbert et al., 2014) and has been applied in models (Wang et al., 2014). In addition, it has been shown that representation of the time evolution of the distribution of contact angles is necessary to improve the representation of ice formation in a cloud-resolving model under some conditions using classical nucleation theory (Savre and Ekman, 2015).

The alternative to describing ice nucleation by classical nucleation theory is to use a singular approximation Vali et al. (2015) in which the time dependence of nucleation is assumed to be of secondary importance compared to the particle-to-particle variability. This approach has been used to define the population of INP in previous model studies (Niemand et al., 2012; Atkinson et al., 2013; Wilson et al., 2015). The ice nucleating efficiency using the singular description is defined by a density of active sites, which is a function of the temperature and usually of the surface area ($n_s$), or another parameter

characteristic of the aerosol population (such as mass or volume) (Murray et al., 2012). This description of ice nucleation is consistent with many laboratory studies showing that particle-to-particle variability is the main factor driving the observed spectrum of INP concentrations with temperature (Vali, 2008; Herbert et al., 2014; Vali and Snider, 2015) for most of the known atmospherically relevant ice nucleating species. However, it should be borne in mind that time dependence could play a role in long-lived stable mixed-phase clouds where ice crystals are produced over a long period of time (Morrison et al.,

2011; Murray et al., 2011; Westbrook and Illingworth, 2013; Herbert et al., 2014). Nevertheless, the singular approach for ice nucleation can be used to approximate INP concentrations, which can be calculated with knowledge of the number, size distribution and density of active sites of the relevant INP species.

    Among the different aerosol species, mineral dust is considered to be the dominant ice nucleating species in many parts of the world (Hoose et al., 2010b; Ardon-Dryer and Levin, 2014; DeMott et al., 2015; Boose et al., 2016). Satellite observations have

shown a negative correlation between the amount of supercooled water and dust concentration (Choi et al., 2010) suggesting that dust might be important for cloud glaciation. The ice nucleating ability of dust has been quantified in a number of studies (Niemand et al., 2012; Broadley et al., 2012; Augustin-Bauditz et al., 2014). Atkinson et al. (2013) found that a mineral component of desert dust, is responsible for most of ice nucleating activity of mineral dust aerosols. Several more recent studies agree with the results shown in (Atkinson et al., 2013) such as (Wex et al., 2014; Harrison et al., 2016; Zolles et al.,

2015; Emersic et al., 2015; O'Sullivan et al., 2014; Niedermeier et al., 2015; Whale et al., 2014). Therefore the representation





of this type of mineral in atmospheric models is important in order to obtain a realistic representation of ice nucleation by mineral dust. We have previously represented ice nucleation on a global scale by K-feldspar aerosols (Wilson et al., 2015; Atkinson et al., 2013). In this study we will take a similar approach to estimate the contribution of K-feldspar aerosol to global INP concentrations.

Phytoplankton and some marine aerosol particles might act as ice nucleating particles. Early evidence for a relationship between phytoplankton and marine INP was found by Schenell and Vali (1975); Schnell and Vali (1976), who observed active INP at temperatures as high as -4$^o C$ in re-suspended biological material, largely from phytoplankton, filtered from bulk sea water. A relationship between the amount of biological material and the concentration of INP was also observed in seawater and fog water by Schnell (1977). More recent studies have observed ice nucleation by *Thalassiosira Psuedonana* (a ubiquitous

species of phytoplankton) diatom cells - (Knopf et al., 2010; Alpert et al., 2011) and exudates (Wilson et al., 2015). However, these studies observed ice nucleation at significantly lower temperatures than those observed by Schnell and Vali (Schenell and Vali, 1975; Schnell and Vali, 1976) suggesting that more active INP could be associated with phytoplankton material in the ocean. This would be supported by a previous observation of ice nucleating bacteria associated with phytoplankton cultures (Fall and Schnell, 1985). Further evidence for the biological origin of marine INP is the heat sensitivity of some types of or-

ganic INP, i.e. the temperature at which they nucleate ice is reduced after heating to 100$^o C$ (Wilson et al., 2015; Schenell and Vali, 1975; Schnell and Vali, 1976) The likelihood of a marine source of INP was highlighted in studies that observed INP concentrations in marine environments remote from other sources of INP (Bigg, 1973; Schnell, 1982; Rosinski et al., 1986; Bigg, 1996; Rosinski et al., 1987, 1988). Using the results from these early studies, Burrows et al. (2013) produced the first global simulation of marine INP concentrations and a comparison with dust INP concentrations, and suggested that marine organics

were likely to dominate the INP population over remote marine regions such as the Southern Ocean. Other studies provide further strong evidence that there is a marine source of atmospheric INP with biological origin. INP production associated with phytoplankton blooms has been observed in laboratory experiments that use artificially generated sea spray aerosol from wave and bubble tanks (Wang et al., 2015; DeMott et al., 2016). DeMott et al. (2016) observed that the INP concentrations in laboratory-generated sea spray were consistent with measurements made by Bigg (1973) as well as with measurements of

ambient INP concentrations in marine-influenced air. Wilson et al. (2015) found that the sea surface microlayer is enriched in INP compared to sub-surface seawater at the same locations. The sea surface microlayer is enriched in surface active organic material similar to that found in sea spray (Cochran et al., 2016; Gantt et al., 2011; Quinn et al., 2014; Aller et al., 2005; Orellana et al., 2011; Russell et al., 2010; Cunliffe et al., 2013). A correlation between total organic carbon content and the temperature at which microlayer droplets froze was observed (Wilson et al., 2015).

All the above evidence suggests the existence of a marine organic source of ice nucleating particles that we will attempt to represent in this paper.

    Here we conduct a modelling study of global immersion mode INP concentrations based on recently developed laboratory-based parameterizations of the ice nucleating ability of two species: marine organic matter and potassium feldspar (K-feldspar). The objectives of our study are to: (i) determine the ability of laboratory-measured INP efficiencies to explain the global

distribution of INP concentrations as a function of activation temperature; (ii) quantify the relative importance of these two





sources of INP in different locations; (iii) determine what fraction of global INP concentrations can be explained by these two major sources and (iv) determine whether, within model and measurement uncertainties, we can use the model results to draw conclusions about additional important sources of INP.

## 2    Methods

### 2.1    Global modelling

We use the GLOMAP-mode global aerosol model described in (Mann et al., 2010). The model has a horizontal latitude-longitude resolution of $2.8^o$x$2.8^o$ and 31 pressure levels from the surface to 10hPa. The species represented in the baseline version are sulphate, sea-salt, black carbon, particulate organic matter and dust. In this study we focus on the representation of two species of relevance to INP: the K-feldspar component of dust and the organic component of primary marine sea spray

aerosols. Aerosol chemical component mass concentrations and the particle number concentration are represented by seven internally mixed log-normal modes (four soluble and three insoluble). Aerosol microphysical processes in the model include nucleation of new particles by gas-to-particle conversion, growth by coagulation and condensation of low-volatility gases, dry deposition at the surface and below-cloud (impaction) and in-cloud (nucleation) wet scavenging. Nucleation scavenging is suppressed for ice clouds (assumed to glaciate at -15$^o$$C$). Scavenging of aerosols by marine drizzle clouds is also included in

the model to improve the predicted concentration in polar regions, as shown in Browse et al. (2012). The model uses wind, temperature and humidity fields from the European Centre for Medium-Range Weather Forecasts (ECMWF). We ran the model from the year 2000 to 2001 in order to reach a steady state aerosol distribution before running the model and then used data from 2001 to 2002.

### 2.2    Representation of feldspar

Feldspar is emitted in the model as a fraction of the mass of dust (derived from AEROCOM emissions (Dentener and Kinne, 2006)). The model has been shown to reproduce dust concentrations accurately (Mann et al., 2010; Huneeus and Schulz, 2011). The fraction of feldspar emitted is assumed to be equal to the fraction by mass of this mineral found in the soils in the arid emission regions. This assumption has been shown to be a close approximation to the fraction of the mineral emitted in the form of aerosols (Lafon et al., 2004; Nickovic et al., 2012). However, new studies suggest that there is a difference between the

fraction of the minerals found in the soil after wet sieving and the aerosolized fraction (Perlwitz et al., 2015). This difference is considered to be small (around a factor of 2) compared to other errors in our representation of the ice nucleating ability of K-feldspar such as differences in the density of active sites of different types of K-feldspar (Harrison et al., 2016).

Feldspar is emitted into the insoluble accumulation and coarse modes with fractions corresponding to the clay and silt size range (Lafon et al., 2004; Nickovic et al., 2012), similar to the method followed in Atkinson et al. (2013). However, once

in the atmosphere, dust particles (including feldspar) are aged by condensation of sulphates and secondary organic aerosol material and moved into the soluble modes, which are subject to wet scavenging. This process was not represented in Atkinson





et al. (2013), and was likely one of the causes of the overestimation of dust concentrations in remote locations as discussed in Atkinson et al. (2013). With this wet scavenging process active, the concentration of feldspar in remote places such as the Southern Ocean is a several orders of magnitude smaller than the concentrations simulated by Atkinson et al. (2013). However, the concentrations closer to source regions are very similar to Atkinson et al. (2013). Compared with other minerals, feldspar

tends to reside in the larger particles as it is found mainly in the silt fraction ($r > 1\mu m$) (Claquin et al., 1999). Due to its size, it is rapidly removed from the atmosphere compared with other mineral species such as those corresponding to the clay fraction as removal by dry deposition increases with particle size. Relatively rapid scavenging of large feldspar-containing particles means that it is transported shorter distances compared with smaller dust particles (that are less rich in feldspar).

### 2.3    Representation of marine organic aerosols

Submicron marine organic aerosols are usually parameterized by relating the organic mass fraction observed in sea-spray to some variables such as seawater chlorophyll content or wind speed (O'Dowd et al., 2015; Rinaldi et al., 2013; Gantt et al., 2011). With those parameterizations, the flux of marine organic mass can be calculated in a model with the flux of sub-micron sea-salt following Eq.A2 (see Appendix) The performance of any parameterization in reproducing observations of marine organic mass concentrations will therefore depend on the emission fluxes of submicron sea-spray, which is a highly

uncertain model-dependent process. Mann et al. (2014) showed that models can have differences of more than a factor of six in the simulated concentration of particles with a diameter larger than 100nm in the Southern Ocean. Other uncertainties affecting the modelled concentrations of marine organic aerosols can arise from removal processes or some other aspect of the model such as the parameterization of convection and cloud microphysical processes, or model grid and temporal resolution. Therefore, the performance of any parameterization in an aerosol model will be affected by the uncertainties related to these

processes. It is therefore necessary to evaluate and adjust the modelled marine organic concentrations to match observations.

To represent primary marine organic aerosols in GLOMAP-mode, we developed a parameterization of the organic mass fraction of submicron sea spray particles to fit the observations of water insoluble organic matter (WIOM) at Amsterdam Island ($37.48^oS$, $77.34^oE$) and Mace Head ($53.33^oN$, $9.9^oW$). We use observations from only these two stations due to the limited availability of long-term measurements of marine WIOM. It is thought that most primary marine organic emissions

are formed of water-insoluble components (Facchini et al., 2008). The marine organic component is assumed to be internally mixed with sea-salt. The sea-salt emissions in our model are dependent on the surface wind speed (10m above the surface) and follow the parameterization of (Gong, 2003), which is an extension of (Monahan et al., 1986). The development of our new organic mass fraction parameterization, explained in detail in Appendix:A, assumes that the organic mass fraction of the sea-spray particles depends on wind speed and the chlorophyll content of seawater. In order to match the seasonal cycle of

WIOM at these two sites, the organic emission parameterization includes a positive dependence of WIOM mass fraction on chlorophyll (O'Dowd et al., 2015; Rinaldi et al., 2013; Gantt et al., 2011), but a negative dependence on wind speed. Thus, the WIOM is essentially diluted in the sea spray particles when the total sea spray emission flux is high, which may be caused by a limited supply of organic material in the surface ocean but effectively limitless salt. This parameterization is similar to previous chlorophyll based parameterizations such as Rinaldi et al. (2013); Gantt et al. (2011) but scaled to fit the observations





in Amsterdam Island and Mace Head. Our model agrees with the observed WIOM concentrations within a factor of two (Fig. 1) which is a small factor compared with other uncertainties related to the calculation of INP concentrations such as the uncertainty related to the parameterization of the number of INP per gram of organic carbon in sea-water (around an order of magnitude) Wilson et al. (2015)

5     The mixed organic-salt sea spray particles are emitted into the accumulation mode and treated as water-soluble particles with respect to their CCN activity, and hence they are removed by nucleation scavenging when they enter a precipitating cloud. This treatment of primary marine organic mass as internally mixed with sea-salt and being able to activate to cloud droplets is consistent with other previous studies (Vignati et al., 2010; Burrows et al., 2013; Orellana et al., 2011; Ovadnevaite et al., 2011; Fuentes et al., 2011; Partanen et al., 2014). Simulated surface concentrations of marine organic aerosol mass are shown

10   in Figure 2.

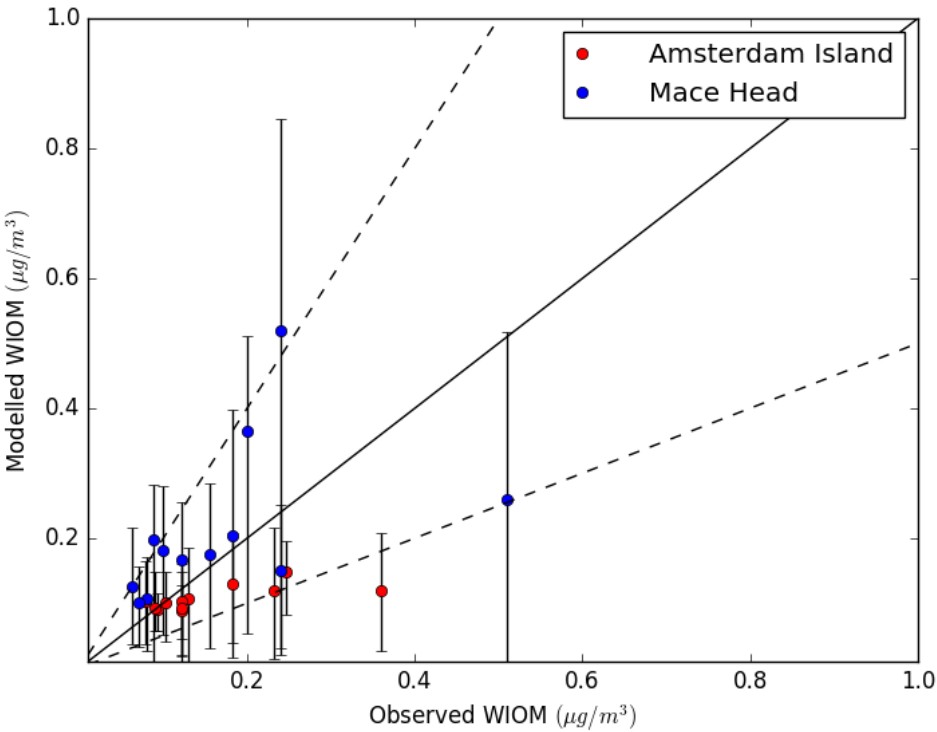

**Figure 1.** Evaluation of modelled WIOM mass concentration with monthly mean observations at Mace Head and Amsterdam Island. The dashed lines correspond to a factor of two difference between modelled and observed values. The error bars correspond to the simulated daily variability within a month (maximum and minimum values). Variability in the observed values is not shown because the measurements were made with filter samples which were collected over 1 week, and therefore they do not represent the day-to-day variability.



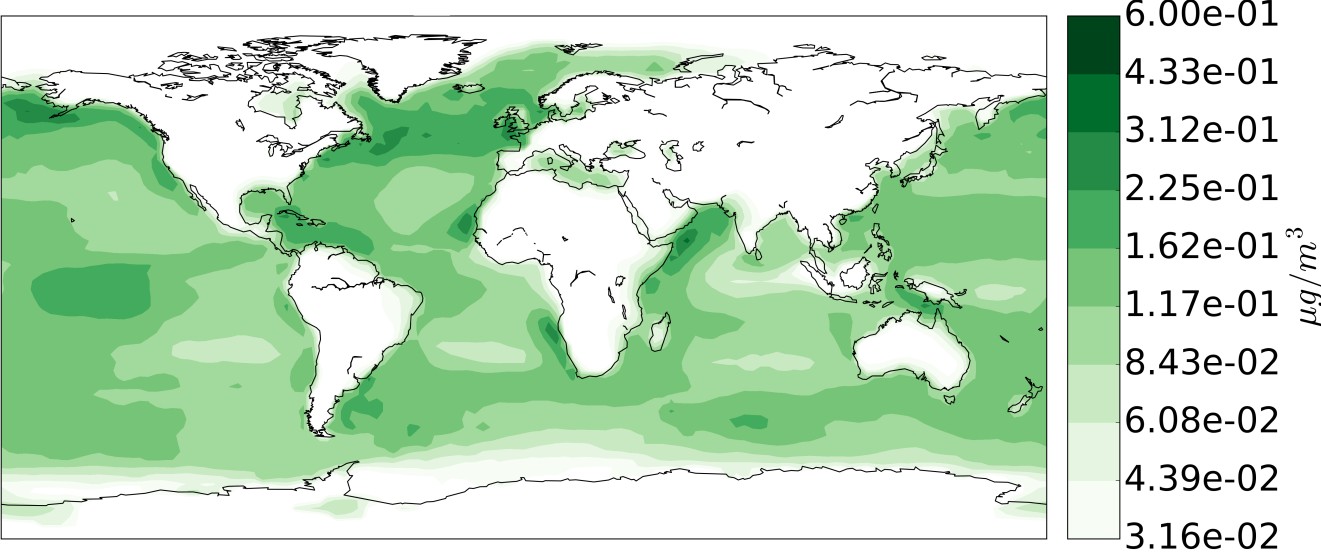

**Figure 2.** Annual mean surface mass concentration of sub-micron marine organic aerosols (WIOM)

## 2.4 Calculation of INP concentrations

To quantify INP concentrations from the modelled aerosol distributions we use the singular description. This method assumes that the time dependence of ice nucleation plays a secondary role and that specific particles have a characteristic temperature at which they nucleate ice. The spectrum of ice nucleating properties is often represented as a surface area density of active

sites dependent on temperature, which is appropriate for solid particles like dust (Atkinson et al., 2013). For marine organic material the active site density is defined per unit mass of organic material in the particle (Wilson et al., 2015).

The method for calculating ice nucleating particle concentrations from the simulated aerosol size distributions is explained in Appendix: B.

To represent the ice nucleating ability of K-feldspar we assume that 35% of the total feldspar is K-feldspar, as assumed in

(Atkinson et al., 2013), then we apply the parameterization for $n_s$ shown in (Atkinson et al., 2013). By using this parameterization we assume that the different varieties of K-feldspar nucleate ice with the same efficiency. Different studies have shown that the values of $n_s$ for most types of potassium feldspar tend to agree with the values shown in (Atkinson et al., 2013) within factor of two to four (Harrison et al., 2016; Emersic et al., 2015; O'Sullivan et al., 2014; Zolles et al., 2015; Niedermeier et al., 2015; Whale et al., 2014). However, it should be borne in mind that a minority of feldspar samples are either much more active

or much less active than indicated by the parameterization defined by (Atkinson et al., 2013). Nevertheless, the Atkinson parameterization is a good approximation of the majority of K-feldspars that have been studied in the laboratory. Assuming that feldspar particles are externally mixed in terms of their mineralogy, we can use the laboratory parameterizations to calculate the INP concentration for each soluble mode, following Eq.B9, as a function of activation temperature (see Appendix:B for the derivation).



For marine organic aerosols, we use the parameterization shown in (Wilson et al., 2015), and apply it to our distributions of simulated marine organic aerosol mass. We are assuming that the organic material found in the sea-surface microlayer is representative of the organic material in sea-spray aerosols and that this material has the same ice nucleating ability as sea-surface microlayer material. For marine organic particles the density of active sites per particle is always small ($\lambda < 0.1$ see Appendix: B) for the whole temperature range covered by the parameterization (-6 to -27$^o C$) and all realistic sizes of particle (submicron particles). This means that we can calculate the INP concentration in a simplified way following Eq.1 (Appendix: B for the derivation).

$$[INP](T) \approx \lambda(T) \cdot [N] \tag{1}$$

It should be noted that extrapolating this parameterization to lower temperatures, or for bigger particles, may lead to unrealistically high concentrations of INP because Eq.1 is no longer valid.

It is important to bear in mind that an INP is defined as a particle which has the potential to nucleate ice if exposed to a specific set of conditions (much like a CCN is defined at a specific super-saturation). For the immersion/condensation mode, the INP concentration we quote are for water saturation and for a defined activation temperature. Hence, there is the question of which activation temperature is most appropriate for displaying the model data. In Figure 4 we illustrate two distinct ways of displaying the model data. The two ways of quoting INP concentrations ($[INP]$, where square brackets indicate concentration) are to quote $[INP]_T$ at a specific activation temperature (T) or to quote $[INP]_{ambient}$ where the activation temperature is set as the local ambient temperature.

In Figure 4a we show the INP concentration for a specific activation temperature ($[INP]_T$) of -20$^o C$ at the 600 hPa pressure level. Throughout much of the globe, especially through the tropics, the temperature at this pressure level will never reach -20$^o C$, so the INP at this altitude that can be active at -20$^o C$ or warmer would not fulfil their potential to nucleate ice. However, if the air at a particular altitude were drawn into a convective system the INP it contains would activate higher in the cloud. Hence, when considering a deep convective cloud where air is moved vertically, it is the spectrum of $[INP]_T$ (a spectrum over activation temperature) which is the pertinent quantity (see Fig. 3 for an illustration). In addition, when comparing measurements of $[INP]$ concentration to our modelled $[INP]$, we compare these quantities at specific activation temperatures. Hence, Figure 4a provides the $[INP]$ to compare to a measurement of $[INP]$ where the activation temperature in a measurement was -20$^o C$.



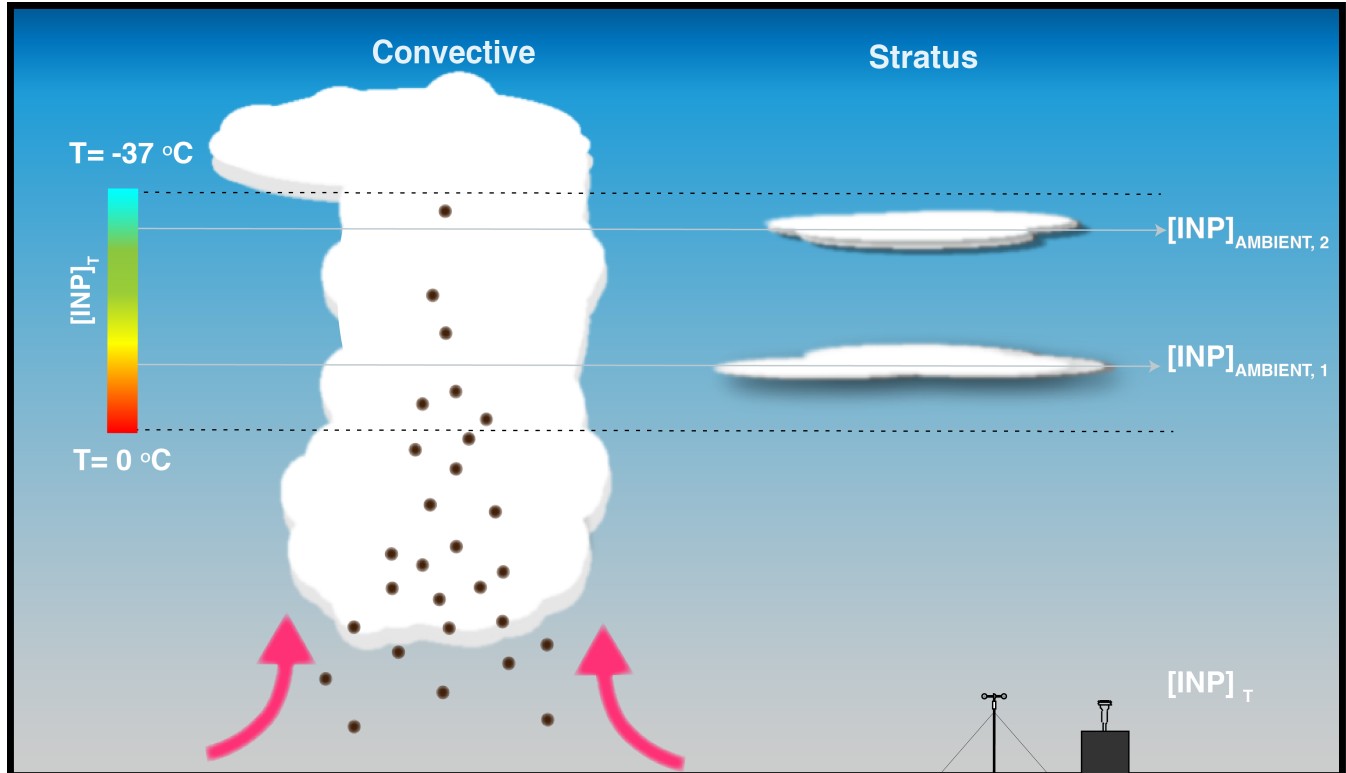

**Figure 3.** Illustration of the two ways in which we display INP concentrations. It is important to bear in mind that INP are defined as particles with the potential to nucleate ice and their concentration is quoted for a specific set of conditions. $[INP]ambient$, where ambient denotes the local atmospheric temperature, is a useful way of looking at the INP concentration relevant to non-deep convective mixed-phase clouds. $[INP]_T$ on the other hand, has utility in representing the spectrum of INP concentrations over temperature that will influence clouds with a large vertical extent such as deep-convective systems. Moreover $[INP]_T$ is the relevant quantity when comparing modelled and observed INP concentrations, since measurements are made by exposing particles to controlled temperatures within the instrumentation.

In Figure 4b we plot the $[INP]$ where the activation temperature is set to the local atmospheric temperature. $[INP]_{ambient}$ is useful to identify regions in the atmosphere where we might expect cloud glaciation in stratus type mixed-phase clouds. Non-deep convective clouds with minimal vertical extent, such as altostratus, altocumulus or high latitude stratus, form in air parcels which have not been vertically transported large distances, in contrast to deep convection. Based on Figure 4b, we would expect K-feldspar to contribute much more to mid-latitude, mid-level (600 hPa), mixed-phase clouds in the Northern Hemisphere than in the Southern Hemisphere.

Both $[INP]_T$ and $[INP]_{ambient}$ are useful ways of looking at the global INP distribution, but in order to understand the impact of these INP species on clouds, we would need a model where the INP fields are coupled to cloud microphysics and dynamics. This is beyond the remit of this study, where our goal is to understand the global distribution of INP.





**Figure 4.** Annual mean K-feldspar INP distribution using GLOMAP-mode at a pressure level of 600hpa. Top panel shows the concentration of ice nucleating particles active at a temperature of -20C ($[INP]_T$) whereas the bottom panel shows the INP concentration at local ambient temperature ($[INP]_{ambient}$).

In order to calculate $[INP]_{ambient}$, we use the daily mean temperatures obtained from ECMWF and the daily mean concentrations (mass and number concentrations) predicted by the model. With these values we can then calculate monthly and annual mean values of INP. The concentrations of $[INP]_{ambient}$ at temperatures colder than the temperature limit of the parameterizations (for K-feldspar: -25$^o$C and marine organics: -27$^o$C) is set at the value defined by the concentration at the limiting temperature of each parameterization. This is consistent with studies that caution against extrapolating singular parameterizations outside the range where measurements were made. For example, Niedermeier et al. (2015) showed that the density of active sites on the surface of K-feldspar particles plateaus below about -25$^o$C and a simple extrapolation of the parameterization of Atkinson et al. (2013) would lead to substantial errors.



## 3 Results

### 3.1 Simulated global INP distributions

Simulated INP concentrations at the surface are shown in Figure 5 for an activation temperature of $-15^oC$. Feldspar dominates the INP concentration in environments influenced by terrestrial dust emission sources such as the Sahara and the Asian dust belt. However, concentrations fall rapidly with distance away from sources because the large size feldspar-containing dust particles are rapidly removed from the atmosphere 5a. The concentrations of INP from K-feldspar and marine organics are summarized in Figure 5c and comparison with panels 5a and 5b reveals that INP from deserts far outnumber INP from sea spray throughout much of the low and mid-latitudes, which are strongly influenced by desert dust, but marine organics become more important over the world's remote oceans, such as the Southern Ocean.







**Figure 5.** Yearly mean distributions of ice nucleating particles concentrations, for an activation temperature of -15$^o C$. Based on feldspar (a) and marine organics (b). (c) shows the total INP concentration obtained by summing the INP concentrations from K-feldspar and marine organics.





Figure 6 shows the $[INP]_{ambient}$ concentration of marine organics and K-feldspar for the different months of the year. Feldspar dominates $[INP]_{ambient}$ on a monthly mean basis across the northern hemisphere, while marine organic aerosols, tend to be important in southern high latitudes, such as those corresponding to the Southern Ocean and Antarctica.

The monthly mean results in Fig.6 have to be interpreted with caution since high dust concentrations are often associated
with episodic dust plumes. Hence, the monthly mean may not reflect the relative contributions of desert dust and sea spray INP on a day-to-day basis. In addition, day-to-day fluctuations in temperature can drive large changes in $[INP]_{ambient}$ which are not necessarily representative of the typical concentrations of active ice nucleating particles, but will greatly affect the monthly mean value of $[INP]_{ambient}$, as the INP concentration increases exponentially with temperature. To account for such variability, Fig.7 shows the percentage of days per season when the concentration of $[INP]_{ambient}$ from marine organics is
greater than the concentration from K-feldspar. Overall, over the northern hemisphere, marine organic INP concentrations are greater than K-feldspar INP concentrations between 10% and 30% of the days when the temperature is within the mixed-phase range and the total concentration of $[INP]_{ambient}$ is larger than $10^{-4} L^{-1}$. This large influence of marine organic INP is hidden when looking at the monthly mean values shown in Figure 6 as the feldspar monthly mean concentrations are dominated by short periods when a dust plume occurs. It is striking that the contribution of marine organics is more important than K-feldspar
on a significant fraction of days in the Northern Hemisphere because in these zonal mean plots we are averaging across the Eurasian and North American continents where the influence of marine organics is minor. In fact, Fig.7 suggest that marine organics are more important than K-feldspar in the North Atlantic, for example, on 10-40 % of days at 600 hPa.

In the Southern Hemisphere, the dominance of marine organic aerosols is more consistent. Both on a monthly mean basis and on the large majority of days, marine organic aerosols are the dominant INP from March through to November (Fig.6b-d).
On the other hand, K-feldspar cannot always be ruled out as an important source of INP in the southern high latitudes in the period from March to November, since there are still several days per month (10 to 60%) when the concentration of transported K-feldspar INP, particularly from South American and Australian sources, dominates over marine organics Fig.6. Conversely, during December to February at southern high latitudes, K- feldspar mineral dust is more important on more days than marine organic aerosols (Fig.7a). This is related to higher dust concentrations during the austral summer.





**Figure 6.** Zonal mean profiles of $[INP]_{ambient}$ for every month of the year. The black contour lines correspond to the INP concentration of K-feldspar aerosols $(m^{-3})$, while the colormap shows the INP concentration of marine organic aerosol The values correspond to monthly mean values calculated using daily concentrations and temperatures.





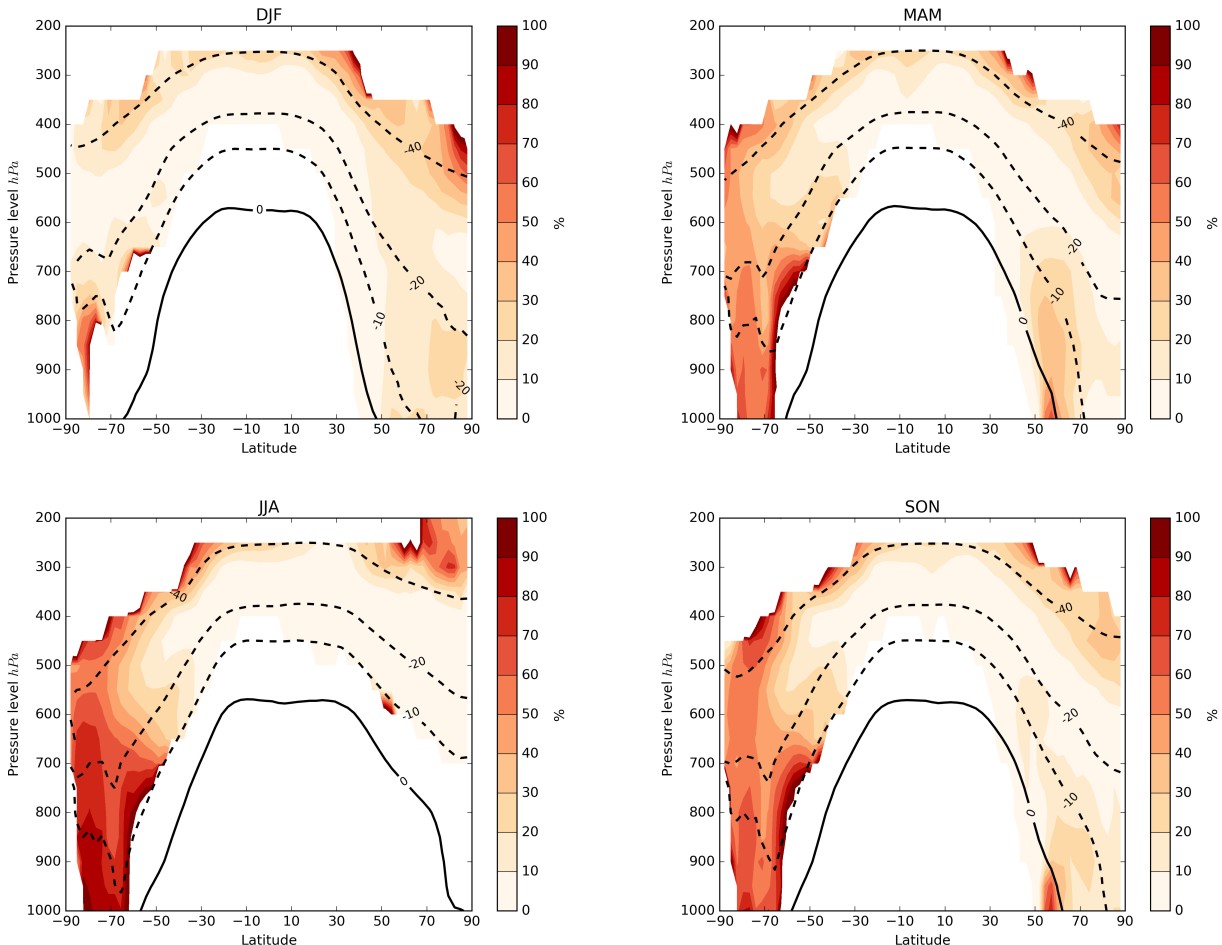

**Figure 7.** Percentage of days when $[INP]_{ambient}$ from marine organic aerosols is greater than from K-feldspar. The number of days have been calculated only for times and locations where the total $[INP]_{ambient}$ concentration is larger than $0.1m^{-3}$. The black contour lines represent isotherms in degrees centigrade.

## 3.2 Comparison with observations and other parameterizations

Some climate models determine heterogeneous freezing using parameterizations that depend only on the temperature, (McCoy et al., 2015a) such as the scheme of Meyers et al. (1992). This type of parameterization does not account for spatial or temporal variations in the aerosol loading and does not differentiate between different aerosol species, both of which actually determine

5  INP concentrations. Other parameterizations such as (DeMott et al., 2010) use empirical evidence from extensive atmospheric measurements to define INP concentrations in terms of the aerosol particle concentration above a defined size.

Such parameterizations implicitly account for the fact that many INP-active species are present in larger particles, such as in dust (Niemand et al., 2012) and biological particles (Tobo et al., 2013). In addition, larger particles are more likely to




carry nano-scale or smaller ice active materials (O'Sullivan et al., 2015). Nevertheless, size-based parameterizations of INP concentration do not account for the source of the particles or differences between marine and terrestrial aerosols, so they may not capture variations and long-term trends.

In Figure 8 we compare several singular INP parameterizations with observations. Panel b compares the observed values of

$[INP]_T$ to those predicted by the scheme of Meyers et al. (1992), which relates $[INP]_T$ to temperature and is independent of aerosol properties. This is clearly a poor representation of many INP measurements in the atmosphere (Table.1). Fig.8c shows the $[INP]_T$ predicted by the parameterization of DeMott et al. (2010), in which $[INP]_T$ is predicted on the basis of the concentration of particles larger than 0.5 $\mu m$ diameter, $n_{aer,0.5}$, and temperature. This parameterization has a similar performance to Meyers et al. (1992) as it still tends to overpredict $[INP]_T$, although scaling of the predicted values by multiplying them

by a factor to fit the observations might greatly improve its performance as it has a better correlation coefficient (Tab.1). We also note that in our analysis we use the annual mean $n_{aer,0.5}$ from our model (without the contribution of sea-salt aerosols), whereas DeMott et al. (2010) used $n_{aer,0.5}$ from measurements coincident with their INP measurements and obtained a better representation of the $[INP]_T$ data (some of which is included in Figure 8).

Fig.8d shows how our model compares with observed $[INP]_T$ using the desert dust parameterization from Niemand et al.

(Niemand et al., 2012) (with no additional marine organic INP). In this case some observations are overestimated by a factor 100-1000, especially those in marine regions (triangles). This overprediction is partly caused by the implicit assumption that all components of dust particle nucleate ice with the same efficiency. Feldspars exist mainly in the large dust particles (silt fraction) so they are not transported as efficiently to remote locations as the clay minerals, consequently transported desert dust is less important as an INP in remote locations.

Finally, we compare our two-species representation of INP with the same $[INP]_T$ dataset Fig.8e-g. The observations used in this comparison are within the range of temperatures of the parameterizations (-5 to -27$^o$C). In this case our representation of INP (8e) is able to reproduce 56.7% of the observations within an order of magnitude and 74% within 1.5 orders of magnitude Tab.1. The contributions of K-feldspar and marine organics to the simulated INP concentrations of each data point are illustrated in Figure 8g. Marine organics explain more than 90% of the INP concentrations in Marine-influenced environments and

some terrestrial environments with low concentrations of INP (corresponding to high temperature observations). K-Feldspar, however, explains most of the observations in terrestrial regions. The large biases observed when using species independent parameterizations over marine regions are largely corrected, as most marine influenced INP concentrations are simulated within an order of magnitude (72% of marine points), although, some biases are apparent. Figure 9 shows the location and temperature of the observations with a bias greater than 1.5 orders of magnitude. Figure 9a suggests that the main positive bias occurs at low

temperatures (<-20$^o$C) in locations far form K-feldspar emission sources, where it is transported. It is possible that processes such as atmospheric aging by acids play a role in modifying the efficiency of K-feldspar aerosols in nucleating ice (Augustin-Bauditz et al., 2014) or that we overestimate the amount of feldspar particles that are transported. One possible explanation for this is that we do not model the preferential removal of INP during cloud glaciation, hence K-feldspar aerosol transported over long distances may contain fewer INP than our model simulations (Stopelli et al., 2015; Haga et al., 2014, 2013). Fig.9b

shows that the model underestimates high-temperature INP concentrations ($\sim -5$ to $-15^oK$) over terrestrial locations, which





| Parameterization | Temperature range | Datapoins | Pt1 | Pt1.5 | R (log) |
|---|---|---|---|---|---|
| Meyers et al. (1992) | $0^oC$ to $-37^oC$ | 479 | 35.5% | 51% | 0.57 |
| DeMott et al. (2010) | $0^oC$ to $-37^oC$ | 479 | 24% | 39.2% | 0.672 |
| Niemand et al. (2012) | $-12^oC$ to $-33^oC$ | 438 | 33.7% | 53% | 0.58 |
| Marine + Kfeldspar | $-6^oC$ to $-25^oC$ | 354 | 56.7% | 74% | 0.625 |

**Table 1.** Statistical performance of the different parameterizations. Pt1 and Pt1.5 are the percentages of datapoints reproduced within an order of magnitude and 1.5 orders of magnitude. The correlation coefficient has been calculated with the logarithm of the values.

might indicate that we are missing some terrestrial source that affects the INP concentration. Some of the possible candidates for these particles could be bacteria (Möhler et al., 2008; Hartmann et al., 2013; Maki and Willoughby, 1978) , fungal material (O'Sullivan et al., 2015, 2016; Fröhlich-Nowoisky et al., 2015; Pouleur et al., 1992; Morris et al., 2013), agricultural dust (O'Sullivan et al., 2014; Tobo et al., 2014; Garcia et al., 2012) or biological nanoscale fragments attached to mineral dust

5     particles (O'Sullivan et al., 2015, 2016; Pummer et al., 2015; Fröhlich-Nowoisky et al., 2015). However, size-resolved INP measurements in several terrestrial locations suggest that a large proportion (40%- 90%) of INP are commonly associated with larger particles (diameter $> 2.5 \mu m$) (Mason et al., 2016). Such large particles are likely to have short atmospheric lifetimes, so they are less likely to be transported to cloud altitudes than smaller particles and are more likely to be transported shorter distances. In summary, the overall agreement between the two-species model and observations is good, but there are significant

10    discrepancies. These discrepancies indicate that processes such as aging and preferential INP in-cloud removal are important and also that we are missing high temperature terrestrial sources of INP in the model.





**Figure 8.** Comparison of the performance of a variety of INP parameterizations tested against field measurements. a)Location of the data used for comparison. (a-g) Modeled INP concentration values when using: b) Meyers parameterization (Meyers et al., 1992) c) DeMott's parameterization (DeMott et al., 2010) combined with a global aerosol simulation using GLOMAP-mode, d) Niemand dust parameterization (Niemand et al., 2012), e) Our two-species based representation based on feldspar (Atkinson et al., 2013) and marine organic aerosols (Wilson et al., 2015). . f) Same as e) but showing the relative contribution (in orders of magnitude) of each aerosol species to the simulated concentration. g) Same as e) but distinguishing between the different campaigns shown in a)(with the same colours and symbols). Triangles represent marine influenced regions and points terrestrial environments. The dashed lines represent one order of magnitude of difference between modelled and observed and the dashed-dotted lines 1.5 orders of magnitude. The simulated values correspond to an annual mean concentration and the error bars correspond to the simulated seasonality of INP calculated with monthly mean values. For Niemand's dust parameterization (Niemand et al., 2012) the range of data is within the range of temperatures shown in Niemand et al. (2012) (-12 to -33$^{o}C$). References to the datasets used are shown in Appendix: C





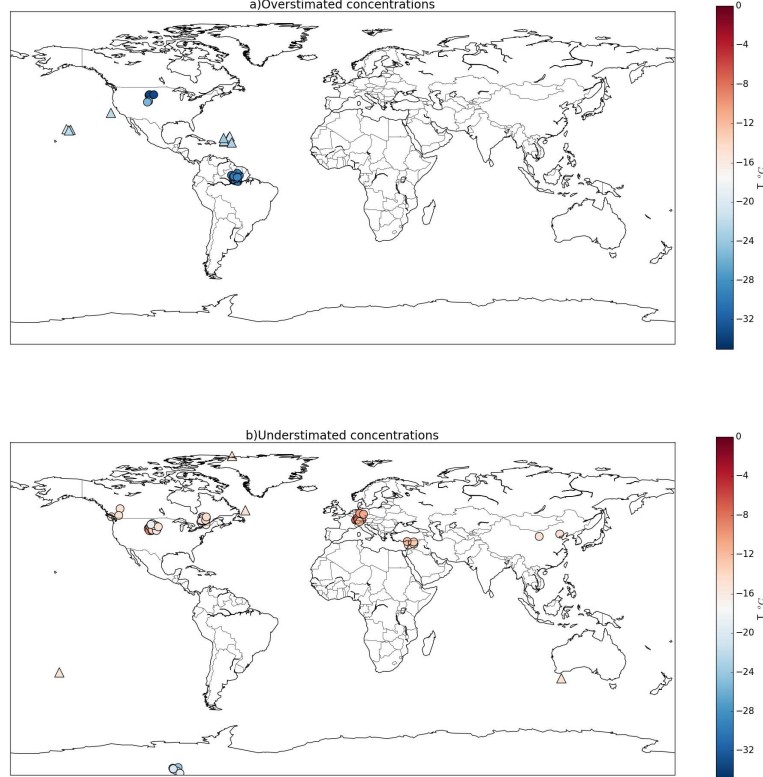

**Figure 9.** Overestimation and underestimation places according to our two species based parameterization of INP (Atkinson et al., 2013; Wilson et al., 2015). a) Shows the places where we overestimate the values of INP by more than 1.5 orders of magnitude. b) Similar to a but for places where the concentration is underestimated by more than 1.5 orders of magnitude. The location of the points have been moved randomly in the plot for purpose of visualization so it can be seen when the bias affect to a single data point or a whole dataset.

## 4 Conclusions

This study is a step towards the inclusion of ice nucleating particles in weather and climate models in a way that accounts for the aerosol chemical composition using laboratory- derived parameterizations under the singular description. By using a representation of INP based on K-feldspar and marine organic aerosols, we can compare the relative importance of these two species. We find that marine organic aerosols dominate the concentration of INP in remote locations like the Southern Ocean on many days, whereas feldspar particles are the dominant species for ice nucleation in places influenced by the terrestrial emission sources.

K-feldspar in our model can reproduce 70% of the observations of INP in terrestrial locations at low temperatures ($T < -15^oC$) within 1.5 orders of magnitude. Because K-feldspar is mainly a coarse aerosol type, it is scavenged more rapidly than





the clay fraction of desert dusts, and therefore has substantially smaller influence on remote marine environments in contrast with Atkinson et al. (2013) where dust was not subject to wet removal. For remote locations, we find that marine organic aerosols acting as INP are able to reproduce a majority (80%) of the observations within an order of magnitude.

Our model of INP based on emitted and transported aerosol species provides a reasonable explanation of measured global INP concentrations, but there are some important biases. The two-species model overestimates by around 1.5 orders of magnitude the concentrations of INP in marine locations that are influenced by the transport of K-feldspar-containing dust particles, although it is difficult to draw firm conclusions from the small number of observations. Nevertheless, the bias points to the possible importance of missing processes, such as the effect of atmospheric processing of feldspar particles, a preferential scavenging of INP as proposed in (Stopelli et al., 2015), or a possible overestimation of the transport of this aerosol type. The model also underestimates measured INP concentrations at high temperatures in some terrestrial locations. This bias is most likely to be explained by neglecting the contribution of some terrestrial biogenic aerosol species such as soil dust, fungal spores and bacteria. The model bias is large at the surface, but some studies show that some of these species are not important for ice nucleation once in the atmosphere (Spracklen and Heald, 2014; Hoose et al., 2010a) because of their low simulated concentrations above the surface for heterogeneous ice nucleation. These species however, could be important for triggering secondary production ice processes, such as the Hallet-Mossop process, due to its high nucleation temperatures. In addition, other unknown sources of ice nucleating particles, such as biological fragments attached to mineral dust particles (O'Sullivan et al., 2015, 2016), could help explain underestimated INP concentrations in the model.

In summary, our results suggest that the inclusion of both marine organic and feldspar emissions are required to accurately simulate global INP concentrations. However, there are still large uncertainties to be resolved, such as the importance of acid coating affecting the INP ability of K-feldspar (Wex et al., 2014; Sullivan et al., 2010) or the relative importance of soot for ice nucleation in the atmosphere, which could lead to a possible anthropogenic effect on clouds.

Finally, we suggest that further experimental studies on the ice nucleating ability of different aerosol species, followed by modelling studies of their importance in the atmosphere, will be crucial for determining the possible importance of other species for ice nucleation under atmospheric conditions. In addition, more measurements in the ambient atmosphere are necessary to better evaluate and constrain models.

## Appendix A: Marine organic emissions

In order to represent the distribution of sub-micron marine organics aerosols, first we simulate the distribution of sea-salt aerosols (SS) with GLOMAP-mode for the year 2001. Then we look at the correlation between the monthly mean emission flux of sea-salt particles in the accumulation mode($100nm < r < 1\mu m$) and the monthly mean surface concentration of sub-micron sea-salt in the grid-boxes corresponding to Mace Head and Amsterdam Island. We then take the grid-boxes that score a correlation $R > 0.9$ and assume that, as a first order approach, the emissions of these grid-boxes will drive the concentrations of sub-micron sea-spray in their corresponding stations (Fig.10). Once these grid-boxes are identified for every station, we calculate the organic mass fraction (OMF) in surface air (lowest model layer) at both stations with modelled concentrations



of sea-spray and measured concentrations of water insoluble organic matter (WIOM) following Eq.A1. The WIOM in Mace Head data is obtained from (Rinaldi et al., 2013) by averaging measurements corresponding to a few days (from 5 to 14 days) in every month. For Amsterdam island, WIOM is derived from (Sciare et al., 2009) using a factor of 1.9 to convert from water insoluble organic carbon to WIOM (Burrows et al., 2013). The chlorophyll-a maps used correspond to monthly mean values

obtained from GLOBCOLOUR (Maritorena and Siegel, 2005), which made use of data from 3 different satellites to merge their chlorophyll-a maps and produce a final product with an enhanced global coverage.

$$OMF = \frac{[WIOM]}{[SS_{mass}] + [WIOM]} \tag{A1}$$

In order to develop a parameterization of the organic mass fraction to be used in both hemispheres we use the monthly mean values of the chlorophyll-a content in the grid-boxes previously related to each station, together with the monthly mean

reanalysis (ECMWF) wind speed at 10 meters over the surface (U10M) of these grid-boxes and relate these two variables to the organic mass fraction previously calculated Fig.11 a). We then fit the OMF to a two-dimensional equation with the wind speed and chlorophyll-a content as variables Fig.11 b). This gives us a parameterization of the OMF emitted with sub-micron sea-spray that can fit our model. In order to avoid unrealistic OMF values due to extrapolation, we limit the maximum value of our OMF to be 0.85.

The mass flux of marine organic material can be then calculated as:

$$Flux_{WIOM} = \frac{Flux_{SS} \cdot OMF}{1 - OMF} \tag{A2}$$



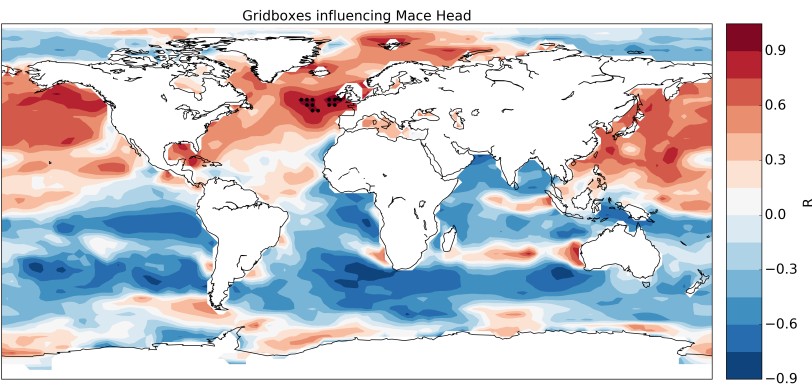

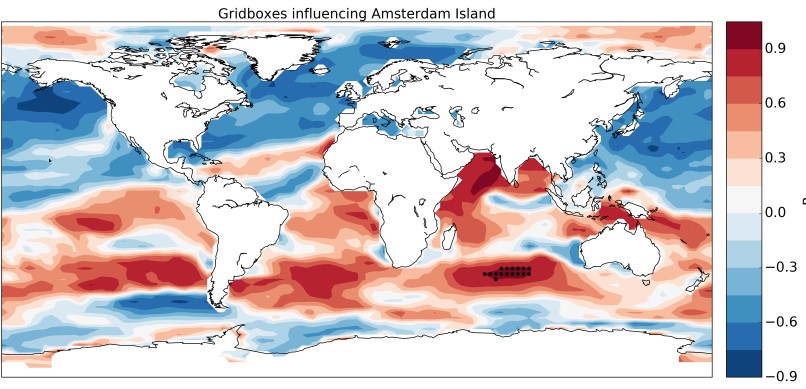

**Figure 10.** Linear correlation values between the monthly emission of sub-micron sea-spray and their monthly concentrations in Top panel: Mace Head, Bottom pannel: Amsterdam Island . The dots represent the grid-boxes that we relate to every station because of having a value R>0.9





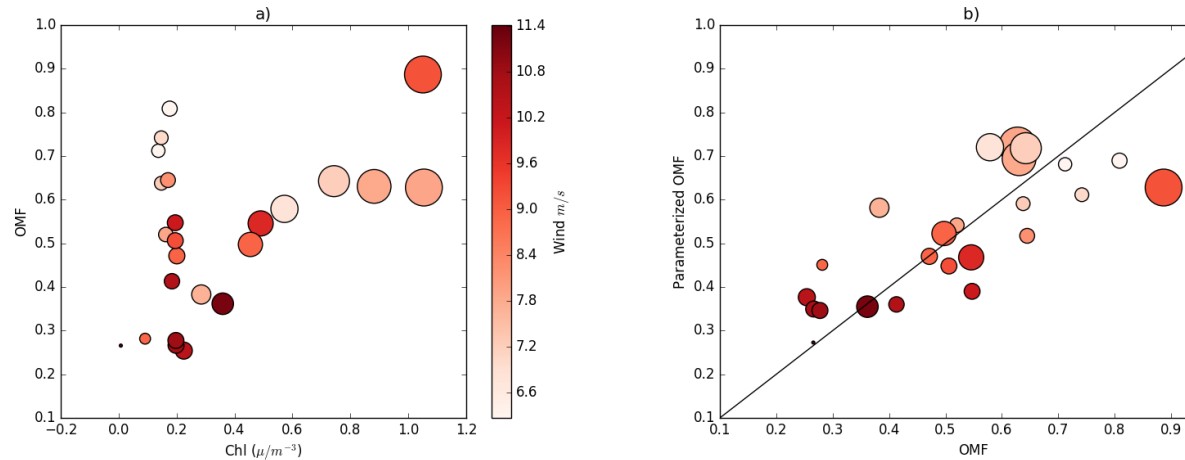

**Figure 11.** OMF compared as a function of chlorophyll-a content and surface wind speed in both stations. The size of the points represent the mean chlorophyll-a content of the grid-boxes related previously to every station (Fig.10), the colour of the points is related to the wind speed of those grid-boxes. The parameterization for the OMF is $OMF = A * [CHL(mg/m^3)] + B * [U10M(m/s)]^C + D$ with $A = 0.241, B = -7.503, C = 0.075, D = 9.274$





## Appendix B: Calculation of INP concentrations

Assuming that the active sites from which ice nucleation can occur under the singular description are randomly distributed in the aerosol population, the probability of one particle to have a certain number of active sites ($k$) can be represented by the poisson distribution Eq.B1.

$$f(k,\lambda) = \frac{e^{-\lambda}\lambda^k}{k!} \tag{B1}$$

Where $f$ is the probability of having $k$ active sites in a particle and $\lambda$ represents the expected value of active sites per particle at a certain temperature ($T$). We can calculate the probability of a particle immersed in a supercooled water droplet to freeze it ($P$), as the sum of the probability of having 1 or more active sites in it Eq. B2:

$$P = \sum_{k=1}^{\infty} f(k,\lambda) \tag{B2}$$

As the sum from $k=0$ to $k=\infty$ of Eq. B1 has to be equal to 1, we can also represent this sum as Eq. B3:

$$P = \sum_{k=0}^{\infty} f(k,\lambda) - f(0,\lambda) = 1 - e^{-\lambda} \tag{B3}$$

If we have a distribution of particles of the same size and same density of active sites, this probability $P$ will be the same for all of them, and so the fraction of supercooled water droplets that will freeze know as fraction frozen ($ff$), will be therefore:

$$ff = 1 - e^{-\lambda} \tag{B4}$$

We can then calculate the INP concentration as:

$$[INP] = ff \cdot [N] \tag{B5}$$

where $[N]$ represent the concentration of a certain type of aerosol. For the case in which we have a density of active sites distributed across the surface area of a particle depending on temperature $n_s(T)$, we can calculate $\lambda$ for a particle of radius $r$ as:

$$\lambda(r,T) = 4\pi r^2 \cdot n_s(T) \tag{B6}$$

Hence:

$$ff(r,T) = 1 - e^{-n_s(T) \cdot 4\pi r^2} \tag{B7}$$





In GLOMAP-mode, the distribution of aerosols is represented in log-normal modes, and their probability density function $PDF$ is given by:

$$PDF(r) = \frac{1}{r \cdot ln(\sigma) \cdot \sqrt{2\pi}} \cdot e^{\frac{-(ln(r)-ln(r_m))^2}{2 \cdot ln(\sigma)^2}} \tag{B8}$$

where $r_m$ is the mean radius of the mode and $\sigma$ the standard deviation of the mode.

The INP concentration is therefore the integral across all the possible values of $r$ for every mode, and it will change for every temperature:

$$[INP]_{mode}(T) = \int_0^\infty (1 - e^{-4 \cdot \pi \cdot r^2 \cdot n_s(T)}) \cdot N \cdot \frac{1}{r \cdot ln(\sigma) \cdot \sqrt{2\pi}} \cdot e^{\frac{-(ln(r)-ln(r_m))^2}{2 \cdot ln(\sigma)^2}} \, dr \tag{B9}$$

In our case, we consider that just the soluble modes can activate into water droplets, so the total INP concentration is the sum of the concentrations for every soluble mode.

In the special case of having a value of $\lambda$ small ($\lambda < 0.1$), we can approximate the value of the fraction frozen ($ff$) using a 1st order Taylor series centred in 0:

$$ff \approx ff_{\lambda=o} + \frac{1}{1!} \frac{\partial ff}{\partial \lambda}\bigg|_{\lambda=0} \cdot \lambda + ... \tag{B10}$$

$$ff_{\lambda=0} = 1 - e^0 = 0 \tag{B11}$$

$$\frac{\partial ff}{\partial \lambda}\bigg|_{\lambda=0} = \left[ -e^{-\lambda} \cdot (-1) \right]_{\lambda=0} \cdot \lambda = 1 \cdot \lambda \tag{B12}$$

$$ff \approx \lambda \tag{B13}$$

In other words, if the number of active sites is small compared with the number of particles, we can approximate the number of particles having one or more actives sites, to the number of active sites. And the INP concentration can be calculated as:

$$[INP](T) \approx \lambda(T) \cdot [N] \tag{B14}$$



| Campaign/dataset | Location | Marine or Terrestrial | Data points | References |
|---|---|---|---|---|
| Bigg73 | Australia | Terrestrial | 24 | (Bigg, 1973) |
| CLEX | East Canada | Terrestrial | 60 | (DeMott et al., 2010) |
| Yin | China | Terrestrial | 21 | (Yin et al., 2012) |
| ICE-L Ambient | Central USA | Terrestrial | 31 | (DeMott et al., 2010) |
| DeMott2016 | Marine locations | Marine | 44 | (DeMott et al., 2016) |
| Conen_JFJ | Jungfraujoch | Terrestrial | 6 | BACCHUS (Conen et al., 2015) |
| Mason2016 terrestrial | Terrestrial locations | Terrestrial | 15 | (Mason et al., 2016) |
| KAD_South_Pole | South Pole | Terrestrial | 8 | BACCHUS (Ardon-Dryer et al., 2011) |
| ICE-L CVI | Central USA | Terrestrial | 27 | (DeMott et al., 2010) |
| Rosisnky | Gulf of Mexico | Marine | 5 | (Rosinski et al., 1988) |
| Bigg1973 | Southern Ocean | Marine | 102 | (Bigg, 1973) |
| Conen_chaumont | Chaumont | Terrestrial | 7 | BACCHUS (Conen et al., 2015) |
| AMAZE-08 | Amazon rainforest | Terrestrial | 63 | (DeMott et al., 2010) |
| INSPECT-I | Central USA | Terrestrial | 13 | (DeMott et al., 2010) |
| Mason2016 Marine | Marine locations | Marine | 6 | (Mason et al., 2016) |
| KAD_Israel | Jerusalem | Terrestrial | 16 | BACCHUS (Ardon-Dryer and Levin, 2014) |
| INSPECT-II | Central USA | Terrestrial | 11 | (DeMott et al., 2010) |

**Table 2.** Table of the datasets used for this study.

## Appendix C: INP dataset

The dataset used in this study is a compilation of published dataset and unpublished data provided by different groups collaborating to the BACCHUS dataset of INP (http://www.bacchus-env.eu/in/index.php). We contacted all the researchers with condensation-immersion freezing INP data advertised in the BACCHUS dataset at the moment of doing this study. The datasets used are listed in table 2. The datasets corresponding to long term measurements in a single location were re-sized to account for a single data point at every temperature. This is done in order to avoid statistical over-weighting of a single location or campaign.



*Acknowledgements.* This study has been funded by the European Union's Seventh Framework Programme (FP7/2007-797 2013) under grant agreement n$^o$ 603445 (BACCHUS), the European Research Council (ERC, 240449 ICE and 648661 MarineIce) and the National Environmental Research Council, (NERC, NE/I013466/1; NE/K004417/1). The global model simulations were performed on the ARCHER UK National Supercomputing Service. We acknowledge Franz Conen and Christoph Huglin who contributed with data used in this paper. Carslaw is a Royal Society Wolfson Merit Award holder. Susannah Burrows was supported by the U.S. Department of Energy, Office of Science (BER). We thank Katty Huang, Louisa Ickes and Ulrike Lohmann for useful discussion about modelling of marine organic aerosols.





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
