# Peer review of "Contribution of feldspar and marine organic aerosols to global ice nucleating particle concentrations"

_Atmospheric Chemistry and Physics, 2016_

## Referee Comment (RC1) · Anonymous Referee #1 · 19 Oct 2016

This paper is a study of using ice nucleation parameterizations accounting for differences in particles ability to nucleate ice. Here they used K-feldspar and marine organic aerosols as ice nucleation particles (INP). They developed a global model of INP concentrations relevant for the mixed-phase clouds based on laboratory and field measurements. They show that simulated INP concentrations compare better with observations when using the two-species parameterization compared to other parameterizations that only consider dust, or temperature. The paper is clear and mostly well written and I think it should be published after addressing a few minor comments.

Minor comments:

-Page 2, lines14-19: I think there are some contradictions in these sentences: The

authors mention the parameterization by DeMott (2010), which is dependent on the INP concentration (greater than a certain size). Then shortly after it is stated that, "However, studies have shown that cloud are sensitive to INP concentrations. " This is already included in the DeMott (2010) parameterization. What DeMott (2010) does not take into account is the variation in nucleation properties (as stated to be an important factor in the previous sentence). I think the sentences on these lines need to be rephrased.

Further, the sentence on line 29-30 is almost identical to the sentence on line 17-18.

-Page 3, line 12. I suggest including references to Marcolli et al. (2007) and Eidhammer et al. (2009), who also included distributions of contact angles in their studies.

Marcolli, C., S. Gedamke, T. Peter, and B. Zobrist (2007), Efficiency of immersion mode ice nucleation on surrogates of mineral dust, Atmos. Chem. Phys., 7(19), 5081– 5091

Eidhammer, T., P. J. DeMott, and S. M. Kreidenweis (2009), A comparison of heterogeneous ice nucleation parameterizations using a parcel model framework, J. Geophys. Res., 114, D06202, doi:10.1029/2008JD011095.

-Page 3, line 15-16 and many other places: There are many citations where the parentheses are misplaced, such as for the Vali et al. (2015) citation. Here it should be ". . ...approximation (Vali et al. 2015) in which the time . . .." Other places, such as page 5, line 6, it should be ". . ...model described in Mann et al. (2010). " Please go trough the manuscript and fix all misplaces parentheses.

-Page 3, line 33. I suggest including reference to Koehler et al. (2010), which also conducted studies of the ice nucleation ability of dust.

Koehler, K. A., Kreidenweis, S. M., DeMott, P. J., Petters, M. D., Prenni, A. J., and Möhler, O.: Laboratory investigations of the impact of mineral dust aerosol on cold cloud formation, Atmos. Chem. Phys., 10, 11955-11968, doi:10.5194/acp-10-11955-2010, 2010.

-Page 3, line 32: I am confused by this sentence: Atkinson et al. (2013) found that a

mineral component of desert dust, is responsible for most of ice nucleating activity of mineral dust aerosols.

Should it be "…...activity of desert dust aerosols"?

-Page 4, line 1: What does "this type of mineral" exactly refer to?

-Page 5, line 13-14: By Nucleation scavenging is suppressed for ice clouds, is it meant that it is not included, meaning that the ice nucleation parameterization is only based on temperature and not INP concentration. This should be explicitly stated. Also, by stating assumed to glaciate at -15C, is it implied that below -15C, the clouds comprise only of solid hydrometeors, and not mixed?

-Page 5, line 21: using "accurately" by stating that the model has been shown to reproduce dust concentration accurately is a strong statement. I suggest rephrasing/rewording.

-Page 9, line 18. Figure 3 has not been mentioned yet. Therefore Figure 4 should be labeled Figure 3 instead. Further, I suggest moving the reference to Fig 3. (which now should be Fig. 4) to a separate sentence, instead of in the parenthesis.. For example: "Figure 3 shows an illustration of the different ways of displaying INP. "

-Page 14, line 1: Are Figure 6 zonal averages?

-Page 14: line 12: Please give a range for the mixed-phased range.

-Page 22, EqA1. I suggest moving Eq.A1 up to line 1, page 22, where the equation is first mentioned.

Technical comments

Page 1 line 2: Replace "of their properties" to "of the cloud properties"

Page 1, line 15: replace "…Southern Ocean at some time of the year" with "…Southern Ocean at some part of the year"
Page 2, line 10: remove "other"

Page 2, line 27: "In future" should be "In the future"

Page 4, line 6: Replace ";" with "and"

Page 5, line 7: Suggest replacing "resolution" with "gridspacing"

Page 6, line 3: Southern Ocean is "a" . Remove "a"

Page 6, line 34: Include: "…..based parameterizations such as in Rinaldi et al. (2013) and Gnatt et al. (2011) but scaled . . ."

Page 8, line 12: I suggest replacing potassium feldspar with K-feldspar for consistency.

Page 12, line 6: 5a should be in parenthesis.

Page 14, line 22: Fig. 6 should be in parenthesis.

Page 22, line 11 and 12. Missing parentheses before Fig.11a and Fig.11 b

Page 25, line 6: Replace Where with Here.

Page 25, line 13: switch : "be therefore" with "therefore be"

---

## Referee Comment (RC2) · Anonymous Referee #2 · 29 Oct 2016

The manuscript "Contribution of feldspar and marine organic aerosols to global ice nucleating particle concentrations" by Vergara-Temprado et al. 2016 investigates the question if an aerosol specific freezing parameterization scheme using feldspar as a terrestrial source of INP and marine organic aerosols as a marine source of INP can better represent the overall global INP distribution in comparison to simple non aerosol specific freezing schemes. They retrieve better results with their scheme compared to earlier easier schemes. Additionally, they investigate the role of the two species used as INP (feldspar and marine organic aerosols). They show that feldspar dominated the global INP population. Nevertheless, they demonstrate that marine organic aerosols are an important INP species, especially over the Southern ocean. The manuscript is

well written. However, some of the analysis needs some further testing and possible improvements. It should be published after major revisions.

Major remarks:

- The argumentation in the abstract and in the introduction is partly not convincing and should be explained a bit clearer or rephrased. Specifically the authors claim that because of a difference in terrestrial and marine INP concentrations, INP species specific parameterization schemes are needed instead of schemes that predominantly stem from terrestrial sources. But if a scheme was developed for terrestrial sources and does not account for marine sources, the INP concentration represented in the model would be different for marine and terrestrial sources as well (it would be smaller above marine regions as shown by the fiels observations). Maybe the argumentation should be split up in two aspects: 1.) Why is it important to account for aerosol species in a freezing parameterization scheme (in general), 2.) Why is it important to also include marine sources? Additionally, the argumentation about the underprediction of the persistance of supercooled clouds oder the Southern ocean and the connection to low INP concentrations (page 2, line 20) could be explained better- is the hyposesis that models overestimate INP concentrations over the Southern ocean which leads to a faster glaciation of the clouds? Can you add references for this hypothesis, e.g. showing an overestimation of INP of models over the Southern ocean?

- Some statements of the singular approximation (in comparison to CNT) sound missleading: you write that the time-dependence is of secondary importance compared to the particle-to-particles variability in case of the singular approximation. When using a simple ns-approach, with one set of fit parameters for one species the particle-to-particle variability is also not really considered. Instead of using an average (single) contact angle for one particle population, an average (single) value for the density of active sites for the particle population is used. I do not see where and how the particle-to-particle variability is better represented in the ns-scheme compared to CNT.

- It is not always clear what kind of model output is used for the analysis. While Fig. 7 seems to be based on daily values, Fig. 8 seems to be calculated using annual means (of n_aer,0.5 and probably also the size of the dust particles for the Niemand scheme). Using annual means for the calculations of the INP concentrations could be meaningless. Freezing is very sensitive to variability in temperature etc.. The INP concentrations should be calculated on a model timestep level and then averaged. If that is already done like this in the manuscript, please explain the methodology better. If it is not done like this, the methodology should be thought through again. It should be shown for one example at least that using annually averages does not influence the result.

- The way the global INP dataset is used and the results are analysed can lead to biases, because it is not used in a uniform way for all parameterization schemes. There are three aspects one could investigate using the dataset, but depending on the aspect the use of the dataset should be different:

1.) Evaluating the parameterization schemes:

To evaluate how well a specific parameterization scheme represent the INP conc. the simulated values should be compared to the observed values only within the valid temperature range of the parameterization scheme. That is what was done in this study. However, that does not tell one how good the parameterization scheme works in a model context where it is used over the whole temperature range (see 3.)).

2.) Comparing the "ability" of the different parameterization schemes within each other:

If one would like to compare how different parameterization schemes compare to each other, the comparison should be done for the same temperature range (in this case the smallest defined temperature range of the parameterization schemes). If they are compared not using the same temperature range it could be that the result does not only show the difference of the parameterization schemes but also other aspects, e.g. one parameterization schemes lacks the INP in high temperature regimes, where another

scheme is not defined (and therefore the $R^2$ is not affected). Using different temperature ranges could lead to a bias towards the scheme with the best defined validity temperature range. E.g. looking at the comparison done in this study, the DeMott et al. 2010 scheme would achieve a much better score if the temperature range between 0 and -4°C would not be taken into account.

3.) Evaluating the model performance:

Finally what is interesting in a model context is how good a specific parameterization scheme is able to represent the global INP concentrations. Also if a parameterization scheme is only defined for a certain temperature range the INP concentration has to be simulated for the whole temperature range. In the presented scheme that means that the INP conc. is 0 above -6°C. If one would like to evaluate the performance of a model using this scheme also the INP conc. above -6°C have to be compared to the simulated one (in this case the simulated conc. being 0).

This manuscript shows aspect Nr. 1, but does not really evaluate the other aspects in a correct way. It is reasonable to define parameterization schemes only for a specific temperature range, but is has to be considered that the schemes are later on in a model context used over the whole temperature range and should give reasonable results for the whole range (also if they are not extrapolated).

Minor remarks and typos:

- Page 1, line 4: Remove space before . .

- Page 2, line 22: "A poor representation ... is important..." sounds missleading.

- Page 2, line 29: Is it proven that freezing is a main model bias?

- One name is missspelled in one citation: Instead of Schenell and Vali 1975, it has to be Schnell and Vali 1975.

- Page 3, line 6: You could add more references here.

- Page 4, line 6: Replace ";" by "and".

- Page 4, line 9: Is it Pseudonana instead of Psuedonana?

- Page 4, line 16. Add . after citation.

- Page 4, line 20: Please state which other studies.

- Page 5, line 2: Skip "major" (you do not know if that are the two major sources).

- Page 5, line 14: What do you mean by saying the clouds are assumed to glaciate at -15°C?

- Page 5, line 27: Please elaborate how large the difference would be in case of different types of feldspar compared to the difference between soil/aerosolized feldspar fraction.

- Page 6, line 3: Remove "a".

- Page 6, line 13: Add . after bracket.

- Page 6, line 18: The OMF parameterization does not cause uncertainty? Or why is this not mentioned?

- Page 6, line 29: It also has physical reasons why WIOM depends pos. on chlorophyl and neg. on wind speed. How you write it, it sounds like this is only due to fitting the observations. Please rephrase and maybe elaborate with 1-2 more sentences.

- Page 7, line 4: Add . after bracket.

- Fig. 1: You could color the errorbars in the same color as the data points to make it easier to differentiate the two locations, especially where WIOM is small.

- Fig. 2: I do not understand the unit of the variable plotted here (or the variable)- is it the accumulated mass of sub-micron marine organics over the whole column?

- Page 8, line 12: Add an "a" after "within".

- Page 9, line 22: Higher in the cloud refers to which temperature? Maybe you could explain that a bit more, it might noch be obvious for every reader.

- Fig. 3: What does the color scale mean next to [INP]_T?

- Page 10, line 1: The reference has to be Figure 4 not 4b.

- Fig. 4: Did you also plot this figure for a different height to check if the picture would then look different? E.g. it could be that the dust distribution is more "present" in the lower figure for a different height. That would be an interesting aspect to look at and mention in the manuscript.

- Fig. 4: Does the lower figure indirectly shows that the temperature in the Arctic is always below -20°C at 600 hPa?

- Page 12, line 3: You should explain why you chose an activation temperature of -15°C, that is quite low for the surface (where you want to simulate the INP conc.).

- Page 12, line 5: Add "dust" in front of "sources".

- Page 12, line 6: Put brackets around "5 a".

- Fig. 5: Does it make sense to use the surface concentration for this plot? Wouldn't it be more reasonable to do the simulations at a higher altitude?

- Fig. 5a: What is the white spot in the plot (bottomleft)?

- Fig 6: Add a label to the colorscale. Which variable is plotted?

- Page 14, line 1, 4 and 5 and caption Fig. 6: You plot seasons and not separate months- adapt the wording.

- Page 14, line 4: Add a space between "Fig." and "6".

- Page 14, line 18: More consistent with what?

- Page 14, line 22: Add brackets around "Fig. 6".

- Fig. 6: It would be more consistent with the following analysis if you would give the INP conc. in 1/l instead of $1/m^3$.

- Fig. 6: Instead of the black contour lines you could also display two plots next to each other, that is maybe better readable. In the second plot the labels of the contour lines are difficult to read (overlap).

- Fig. 7: Why do you have values in the temperature range below $-26°C$?

- Fig. 7: Especially in the third plot there are INP values even below $-40°C$- you should explain these "artefacts" or whatever it is.

- Caption Fig. 7: Add a space between label "ambient" and "concentration" (line 2).

- Page 17, line 1: Other schemes indirectly capture the source since large particles sediment and are more predominant close to the source region. Why is only a species-differentiating scheme able to capture variations and long-term trends?

- Page 17, line 6: Add a space between "Table" and "1" (remove the . or write Tab.). Add a space between "Fig." and "8c".

- Page 17, line 10: There is no improvement shown in Tab. 1 (the unscaled values or not shown)? Eather add it in the table, or remove the reference to the table.

- Page 17, line 23: Add brackets around "Tab. 1".

- Table 1: Why is the correlation coefficient calculated for the logarithm of the values? Please explain shortly in the manuscript.

- Page 18, line 10: Since you do not know if preferential INP in-cloud removal is important you should change "are" to "could be". Same in line 11 for the terrestrial source of INP.

- Fig. 8 f is not mentioned in the text. Is this figure necessary? It would need some further explanation to be easy understandable.

- Fig. 8 and Fig. 9 and Fig. 10: Axis labels etc . are quite small font.

- Fig. 8: Label b is truncated.

- Fig. 8 label: Add which simulated and observed variable it is.

- Fig. 8 caption, line 5: Remove one ".".

- Fig. 8 caption, last line: Add a ".".

- Fig. 9 caption, line 2: Add a bracket after "a".

- Page 21, line 24: What kind of measurements would be needed? It would be helpful to elaborate that in 1-2 more sentences.

- Page 22, line 26: Please explain this formula a bit more.

- Fig. 11: Are that yearly mean values or for which time period is the comparison/relation plotted?

- Fig. 11 b) is not explained.

- Fig. 11, caption: add space between the fit parameters. Add a "." at the end of the caption.

- Appendix B: How do you get from Eq. B2 to B3?

- Page 26, line 1: Do you refer to size distribution when you write "distribution"?

- Appendix B: What does the last section mean in your model context?

- Table 2: Remove brackets around the references.

- Table 2: Are the references unpublished where you did add the label "BACCHUS"? Otherwise I do not understand why this is labeled like this and what it means.

General remarks:

- The citations are not done consistent- sometimes brackets are used where there

should not be, sometimes brackets are missing, e.g. at page 3 line 15 brackets are missing vs. at page 2 line 34 brackets should be removed. Please thoroughly go through the citations again.

- Please add a space between numbers and units, e.g. page 5 line 7: 10 hPa.

- Units should not be italic, e.g. page 6 line 5.

- The naming of the modell is not consistent throughout the paper, sometimes you write GLOMAP, sometimes GLOMAP-mode. This should be explained (if the names are different on purpose) or made consistent.

- Reduce the space between the single letters within your variables INP and ff, that increases the readability.

- Be consistent with writing OMF as a variable in italic or not, e.g. page 24, line 3.

---

## Author Comment (AC1) · 21 Dec 2016

Jesús Vergara Temprado

[revised manuscript text omitted]

---

## Author Comment (AC2) · 21 Dec 2016

Response to comment #1

We would like to thank the reviewer for their useful and constructive comments. Our response and the subsequent modifications to the paper are structured as follows:

 *Blue for the reviewer comment*

Normal text for our answers

**Bold for the changes in the manuscript**

*-Page 2, lines14-19: I think there are some contradictions in these sentences: The authors mention the parameterization by DeMott (2010), which is dependent on the INP concentration (greater than a certain size). Then shortly after it is stated that, "However,studies have shown that cloud are sensitive to INP concentrations. " This is already included in the DeMott (2010) parameterization. What DeMott (2010) does not take into account is the variation in nucleation properties (as stated to be an important factor in the previous sentence). I think the sentences on these lines need to be rephrased.*

*Further, the sentence on line 29-30 is almost identical to the sentence on line 17-18.*

We have rephrased these lines to clarify that these parameterizations currently do not represent the differences in the ice nucleating properties of different aerosol species. We also deleted the last sentence.

**The current representation of heterogeneous freezing in climate models and operational numerical weather prediction models is usually based on parameterizations that depend on the temperature (Young et al. 1974, Meyers et al. 1992) or the size distribution of aerosol particles as well as the temperature (DeMott et al. 2010). These parameterizations treat aerosol particles all around the globe and across seasons as having the same ice-nucleating properties irrespective of the aerosol chemical composition. Representing these differences may lead to a better simulation of INP concentrations, thereby improving the representation of mixed-phase clouds.**

*-Page 3, line 12. I suggest including references to Marcolli et al. (2007) and Eidhammer et al. (2009), who also included distributions of contact angles in their studies.*

References added.

*-Page 3, line 15-16 and many other places: There are many citations where the parentheses are misplaced, such as for the Vali et al. (2015) citation. Here it should be "..approximation (Vali et al. 2015) in which the time …" Other places, such as page 5, line 6, it should be "…model described in Mann et al. (2010). " Please go trough the manuscript and fix all misplaces parentheses.*

Done

*-Page 3, line 33. I suggest including reference to Koehler et al. (2010), which also conducted studies of the ice nucleation ability of dust.*

Done

*-Page 3, line 32: I am confused by this sentence: Atkinson et al. (2013) found that a mineral component of desert dust, is responsible for most of ice nucleating activity of mineral dust aerosols. Should it be "..activity of desert dust aerosols"?*

This has been rephrased to: **Atkinson et al. (2013) found that K-feldspars are far more effective at nucleating ice than any of the other minerals in desert dust.**

*Page 4, line 1: What does "this type of mineral" exactly refer to?*

Refers to K-feldspar, we have rephrased this sentence

**Therefore the representation of K-feldspar in atmospheric models…**

*-Page 5, line 13-14: By Nucleation scavenging is suppressed for ice clouds, is it meant that it is not included, meaning that the ice nucleation parameterization is only based on temperature and not INP concentration. This should be explicitly stated. Also, by stating assumed to glaciate at -15C, is it implied that below -15C, the clouds comprise only of solid hydrometeors, and not mixed?*

This refers to the model assumptions that we need to make in order to represent nucleation scavenging of aerosol particles in our model. As we are using a chemical transport model, aerosols do not interact with clouds in any way (other than being scavenged by precipitation) and all the meteorological fields (including cloud fields) come from ECMWF reanalysis. With these fields, we can predict the concentrations of different aerosol species and from the concentrations we calculate offline (after the model simulation) the INP concentrations. The discussion of the nucleation scavenging assumptions is included in Browse et al (2012), so we refer to it for a more detailed description.

**Nucleation scavenging is suppressed for ice clouds, which are assumed to glaciate at -15ºC. A discussion of the nucleation scavenging assumptions in our model is included in Browse et al. (2012).**

-Page 5, line 21: using "accurately" by stating that the model has been shown to reproduce dust concentration accurately is a strong statement. I suggest rephrasing/rewording.

Reworded to 'within an order of magnitude'

**The model has been shown to reproduce dust mass concentrations within an order of magnitude**

*-Page 14, line 1: Are Figure 6 zonal averages?*

Yes. Clarification has been added to figure caption

*-Page 14: line 12: Please give a range for the mixed-phased range.*

Done, added: "(0°C to -37°C)"

*-Page 22, EqA1. I suggest moving Eq.A1 up to line 1, page 22, where the equation is first mentioned.*
Done

*Technical comments*
*Page 1 line 2: Replace "of their properties" to "of the cloud properties"*
Done

*Page 1, line 15: replace "*
*. . .*
*Southern Ocean at some time of the year" with*
*"*
*. . .*
*Southern Ocean at some part of the year"*
Done

*Page 2, line 10: remove "other"*
Done

*Page 2, line 27: "In future" should be "In the future"*
Done

*Page 4, line 6: Replace ";" with "and"*
*Done*

*Page 5, line 7: Suggest replacing "resolution" with "gridspacing"*
*Done*

*Page 6, line 3: Southern Ocean is "a" . Remove "a"*
*Done*

*Page 6, line 34: Include: "*

*. . .*

*..based parameterizations such as in Rinaldi et al. (2013)*
*and Gnatt et al. (2011) but scaled*

*. . .*

*"*

Done

*Page 8, line 12: I suggest replacing potassium feldspar with K-feldspar for consistency.*
Done
*Page 12, line 6: 5a should be in parenthesis.*
Done
*Page 14, line 22: Fig. 6 should be in parenthesis.*
Done

*Page 22, line 11 and 12. Missing parentheses before Fig.11a and Fig.11 b*
Done

*Page 25, line 6: Replace Where with Here.*
Done

*Page 25, line 13: switch : "be therefore" with "therefore be*

Done

---

## Author Comment (AC3) · 21 Dec 2016

Response to comment #2

We would like to thank the reviewer for their useful and constructive comments. Our response and the subsequent modifications to the paper are structured as follows:

 *Blue for the reviewer comment*

Normal text for our answers

**Bold for the changes in the manuscript**

*- The argumentation in the abstract and in the introduction is partly not convincing and should be explained a bit clearer or rephrased. Specifically the authors claim that because of a difference in terrestrial and marine INP concentrations, INP species specific parameterization schemes are needed instead of schemes that predominantly stem from terrestrial sources. But if a scheme was developed for terrestrial sources and does not account for marine sources, the INP concentration represented in the model would be different for marine and terrestrial sources as well (it would be smaller above marine regions as shown by the field observations). Maybe the argumentation should be split up in two aspects: 1.) Why is it important to account for aerosol species in a freezing parameterization scheme (in general), 2.) Why is it important to also include marine sources?*

The referee's logic is correct, up to the point where they state "*But if a scheme was developed for terrestrial sources and does not account for marine sources, the INP concentration represented in the model would be different for marine and terrestrial sources as well (it would be smaller above marine regions as shown by the field observations).*"  This is not correct.  The parameterisations in the literature we refer to simply treat all aerosol identically in all locations or simply assume a temperature dependent concentration of INP. Hence, for example, the Meyers scheme 'predicts' the same INP spectrum over the ocean as over the land. We stress, that we have developed a global model of INP concentrations using a global aerosol model, hence can link aerosol specific INP properties to specific aerosol species.

*Additionally, the argumentation about the underprediction of the persistance of supercooled clouds oder the Southern ocean and the connection to low INP concentrations (page 2, line 20) could be explained better- is the hyposesis that models overestimate INP concentrations over the Southern ocean which leads to a faster glaciation of the clouds? Can you add references for this hypothesis, e.g. showing an overestimation of INP of models over the Southern ocean?*

A discussion of this hypothesis is included in DeMott et al. (2016). Additionally Figure 8 (a,b,c) shows an overestimation of INP in marine environments (triangles) when using 3 commonly used parameterizations.

In order to clarify the arguments here we have extended the discussion:
"Over the Southern Ocean clouds tend to persist in a supercooled state more commonly than models predict (Bodas-Salcedo et al., 2014), which might be related to very low INP concentrations in this region."

To:

**Over the Southern Ocean clouds tend to persist in a supercooled state more commonly than models predict (Bodas-Salcedo et al., 2014), which might be related to very low INP concentrations in this region (Bigg et al. 1973; DeMott et al. 2016). It has been shown that less INP in the Southern Ocean lead to less ice and more supercooled water in model clouds, with a significant impact on the radiative properties of the clouds (Tan et al. 2016).**

*- Some statements of the singular approximation (in comparison to CNT) sound missleading: you write that the time-dependence is of secondary importance compared to the particle-to-particles variability in case of the singular approximation. When using a simple ns-approach, with one set of fit parameters for one species the particle-to-particle variability is also not really considered. Instead of using an average (single) contact angle for one particle population, an average (single) value for the density of active sites for the particle population is used. I do not see where and how the particle-to-particle variability is better represented in the ns-scheme compared to CNT.*

The referee is incorrect in the statement that "average (single) value for the density of active sites for the particle population is used". The parameterisation is a temperature dependent function describing the cumulative density of active sites which become active on decreasing temperature. In this model, a specific site has a characteristic temperature at which it nucleates ice and each particle has a distinct population of active sites. If we applied an average density of active sites, then all particles that possessed that site would trigger freezing at the same temperature. The cited experimental work shows that materials have a spectrum of nucleation sites. Assuming a single contact angle also means that each particle (of the same size) has the same probability of nucleating ice and according to classical theory, nucleation will occur over a narrow range of temperatures. Experimental work suggests that this is not the case and that there is a distribution of sites.

A very important distinction between using a single contact angle parameterisation and the singular description is that when using classical nucleation theory with a single contact angle, eventually all your aerosol particles will freeze, as time dependence is the main factor that drives nucleation. That is in contrast with the singular description and many laboratory studies that show how just a fraction of particles nucleate. This topic has been widely discussed in the past so, and we have referred the interested reader to Appendix 2 and the references therein for a more detailed discussion.

We have improved the clarity of the discussion here by changing: "The ice nucleating efficiency using the singular description is defined by a density of active sites, which is a function of the temperature and usually of the surface area (ns),"

To **"The ice nucleating efficiency using the singular description is defined by a temperature dependent density (i.e. per unit surface area) of active sites, (ns(T) which represents a spectrum of active sites with variable characteristic ice nucleation temperatures. The temperature dependent number of active sites can also be normalised to another parameter characteristic of the aerosol population (such as mass or volume) (Murray et al., 2012). From this density of active sites, one can calculate what fraction of the particles will nucleate ice at a certain temperature (See Appendix:2 )"**

*- It is not always clear what kind of model output is used for the analysis. While Fig. 7 seems to be based on daily values, Fig. 8 seems to be calculated using annual means (of n_aer,0.5 and probably also the size of the dust particles for the Niemand scheme). Using annual means for the calculations of the INP concentrations could be meaningless. Freezing is very sensitive to variability in temperature etc.. The INP concentrations should be calculated on a model timestep level and then averaged. If that is already done like this in the manuscript, please explain the methodology better. If it is not done like this, the methodology should be thought through again. It should be shown for one example at least that using annually averages does not influence the result.*

Figure 7 was calculated using daily values of the temperature and concentration in order to account for the large temperature dependence of the simulated INP concentrations. Figure 8, however, does not depend on the modelled temperatures, as the temperature used to calculate INP is that corresponding to each observation and this is independent of the ambient local temperature. In order words, for an observation at a temperature T1, we calculated the predicted INP concentration with the annual mean concentrations of aerosols given by our model and the temperature at which the observation was done (T1).
We have modified the figure caption in order to clarify this concept.

**For each individual observation, we calculated the INP concentration at the temperature corresponding to the temperature that aerosol particles were exposed to in the INP instruments.**

*- The way the global INP dataset is used and the results are analysed can lead to biases, because it is not used in a uniform way for all parameterization schemes. There are three aspects one could investigate using the dataset, but depending on the aspect the use of the dataset should be different:*
*1.) Evaluating the parameterization schemes:*
*To evaluate how well a specific parameterization scheme represent the INP conc. the simulated values should be compared to the observed values only within the valid temperature range of the parameterization scheme. That is what was done in this study. However, that does not tell one how good the parameterization scheme works in a model context where it is used over the whole temperature range (see 3.)).*
*2.) Comparing the "ability" of the different parameterization schemes within each other:*
*If one would like to compare how different parameterization schemes compare to each other, the comparison should be done for the same temperature range (in this case the smallest defined temperature range of the parameterization schemes). If they are compared not using the same temperature range it could be that the result does not only*

*show the difference of the parameterization schemes but also other aspects, e.g. one parameterization schemes lacks the INP in high temperature regimes, where another scheme is not defined (and therefore the $R_2$ is not affected). Using different temperature ranges could lead to a bias towards the scheme with the best defined validity temperature range. E.g. looking at the comparison done in this study, the DeMott et al. 2010 scheme would achieve a much better score if the temperature range between 0 and -4◦C would not be taken into account.*

*3.) Evaluating the model performance:*
*Finally what is interesting in a model context is how good a specific parameterization scheme is able to represent the global INP concentrations. Also if a parameterization scheme is only defined for a certain temperature range the INP concentration has to be simulated for the whole temperature range. In the presented scheme that means that the INP conc. is 0 above -6◦C. If one would like to evaluate the performance of a model using this scheme also the INP conc. above -6◦C have to be compared to the simulated one (in this case the simulated conc. being 0).*

*This manuscript shows aspect Nr. 1, but does not really evaluate the other aspects in a correct way. It is reasonable to define parameterization schemes only for a specific temperature range, but is has to be considered that the schemes are later on in a model context used over the whole temperature range and should give reasonable results for the whole range (also if they are not extrapolated).*

We have done some changes to address this comment.

First, we have included in figure 8 the datapoints outside the temperature range of the different parameterizations with semitransparent markers. This is done in order to have a visual comparison of how the parameterizations will look like if they are extrapolated.

Then, we have added in Table 1 the same statistical values as before, but also for the other two aspects (all the temperature range and just for the shared temperature range).

With these changes, we think that the 3 main aspects are addressed. Overall. The changes are very minor to the plots, with relatively few data points being added.
We added in the text:

**When the parameterizations are extrapolated outside their temperature range, they still perform similarly.**

**Looking at the performance of the different ways of representing INP within the smallest temperature range shared by the all the parameterizations (-12 to -25C), our representation of INP is able to reproduce 61.6% of the datapoints within and order of magnitude and 78.7% within 1.5 orders of magnitude. These values are greater than the obtained when using the other 3 parameterizations used for this study (Table.1 )**

The caption of Table 1 has changed as well:

**Statistical performance of the different parameterizations. Pt1 and Pt1.5 are the percentages of datapoints reproduced within an order of magnitude and 1.5 orders of magnitude in the temperature range of every parameterization. The number of datapoints used for calculating these values is shown under the 'Datapoints' column. The values with * show the same calculation but including**

**datapoints outside the temperature range of the parameterizations. These values give an idea of the performance that you would expect if you extrapolate the parameterizations in a climate model. The values with ** are for datapoints within the smallest temperature range shared by the 4 parameterizations (-12C to -25C). The correlation coefficient has been calculated with the logarithm of the values as INP concentrations vary logarithmically with temperature.**

*Minor remarks and typos:*

*- Page 1, line 4: Remove space before . .*
Done

*- Page 2, line 22: "A poor representation ... is important..." sounds missleading.*
Modified to:

**A better representation of mixed-phase clouds in climate models has been shown to be important for climate prediction.**

*- Page 2, line 29: Is it proven that freezing is a main model bias?*
It has been proven that freezing is poorly represented in climate models (see McCoy et al., 2015 figure 1). This reference is given in the text.

*- One name is missspelled in one citation: Instead of Schenell and Vali 1975, it has to be Schnell and Vali 1975.*
Corrected

*- Page 3, line 6: You could add more references here.*
Added Sesartic et al (2013) and Lohmann et al. (2006)

*- Page 4, line 6: Replace ";" by "and".*
Done
*- Page 4, line 9: Is it Pseudonana instead of Psuedonana?*
Yes, corrected

*- Page 4, line 16. Add . after citation.*
Done

*- Page 4, line 20: Please state which other studies.*
The other studies are cited in the following sentences, I have rephrased the sentence to connect it with the following sentences.
**Further evidence for the biological origin of marine INP is the heat sensitivity of some types of organic INP, i.e. the temperature at which they nucleate ice is reduced after heating to 100C  (Wilson et al. 2015, Schnell et al. 1975, Schnell et al. 1976).**

*- Page 5, line 2: Skip "major" (you do not know if that are the two major sources).*
Done

*- Page 5, line 14: What do you mean by saying the clouds are assumed to glaciate at 15∘C?*

As we are using a chemical transport model, all the meteorological fields are obtained from ECMWF, including clouds, and our aerosols do not feedback in clouds. Because of this reason, we have to assume a temperature for representing in-cloud scavenging of aerosol particles in ice and liquid clouds. A more detailed evaluation of this assumption as well as a detailed description of the in-cloud scavenging scheme is described in Browse et al. (2012).
We have inserted:
**'A discussion of the nucleation scavenging assumptions in our model is included in Browse et al. (2012)'**

*- Page 5, line 27: Please elaborate how large the difference would be in case of different types of feldspar compared to the difference between soil/aerosolized feldspar fraction.*

Most k-feldspar samples have ice nucleating abilities that agree with each other within a factor of 6. This factor is substantially larger than a factor of 2.

**…ice nucleating ability of K-feldspar such as differences in the density of active sites of different types of K-feldspar (around a factor of 6) (Harrison et al. (2016).**

*- Page 6, line 3: Remove "a".*
Done

*- Page 6, line 13: Add . after bracket.*
Done

*- Page 6, line 18: The OMF parameterization does not cause uncertainty? Or why is this not mentioned?*
Added a comment on the OMF uncertainty

**…processes, or model grid and temporal resolution, as well as uncertainties related to the organic mass fraction parameterization.**

*- Page 6, line 29: It also has physical reasons why WIOM depends pos. on chlorophyl and neg. on wind speed. How you write it, it sounds like this is only due to fitting the observations. Please rephrase and maybe elaborate with 1-2 more sentences*

A more in deep explanation of the dependence of WIOM with windspeed is included in Gantt et al (2011) (figure 1). We have rephrased this section to clarify

**The development of our new organic mass fraction parameterization, explained in detail in Appendix A , assumes that the organic mass fraction of the sea-spray particles depends on wind speed and the chlorophyll content of seawater. The organic emission parameterization includes a positive dependence of WIOM mass fraction on chlorophyll (O'Dowd et al., 2015; Rinaldi et al., 2013; Gantt et al., 2011), but a negative dependence on wind speed.Thus, the WIOM is essentially diluted in the sea spray particles when the total sea spray emission flux is high, which may be caused by a limited supply of organic material in the surface ocean but effectively limitless salt (Gantt et al 2011). This parameterization is similar to previous chlorophyll based parameterizations such as Rinaldi et al.(2013) and Gantt et al (2011) but scaled in order to fit the observations in Amsterdam Island and Mace Head when applied in our model.**

.
*- Page 7, line 4: Add . after bracket.*
Done
*- Fig. 1: You could color the errorbars in the same color as the data points to make it easier to differentiate the two locations, especially where WIOM is small.*
Done

*- Fig. 2: I do not understand the unit of the variable plotted here (or the variable)- is it the accumulated mass of sub-micron marine organics over the whole column?*
No, it refers to the concentration of sub-micron marine organic aerosol mass at the surface. We have changed the description
**Annual mean mass concentration of sub-micron marine organic (µg m$^{-3}$) aerosol at surface level**

*- Page 8, line 12: Add an "a" after "within".*
Done

*- Page 9, line 22: Higher in the cloud refers to which temperature? Maybe you could explain that a bit more, it might noch be obvious for every reader.*
Added the range of temperatures
***Hence, when considering a deep convective cloud where air is moved vertically through all the mixed-phase range of temperatures…***

*- Fig. 3: What does the color scale mean next to [INP]_T?*
It is an example color scale referring to temperatures decreasing from 0 to -37C.

*- Page 10, line 1: The reference has to be Figure 4 not 4b.*
Changed from Figure 4b to Figure 4 (bottom)

*- Fig. 4: Did you also plot this figure for a different height to check if the picture would then look different? E.g. it could be that the dust distribution is more "present" in the lower figure for a different height. That would be an interesting aspect to look at and mention in the manuscript.*
Figure 5a shows a similar picture but for surface level. A comparison of the influence of both marine organic and K-feldspar for all heights is done in figures 6 and 7.

*- Fig. 4: Does the lower figure indirectly shows that the temperature in the Arctic is always below -20∘C at 600 hPa?*
No, the bottom panel shows the annual mean INP concentration active at local ambient temperature.

*- Page 12, line 3: You should explain why you chose an activation temperature of -15∘C, that is quite low for the surface (where you want to simulate the INP conc.).*
It is a temperature at which many atmospheric observations of INP are made, so they could be compared with what this paper predicts. We stress, that [INP]$_{15}$ is independent of local ambient temperature.

In the figure caption we have added a statement:

**We show [INP]$_T$ for a *T* of -15°C because this is a temperature used by many instruments. The number of INPs that activate to ice crystals ([INP]$_{ambient}$) at the surface will be zero over much of the globe, because these particles will only**

**become important at high altitudes. Surface concentrations are show because this is where most observations of atmospheric INP concentrations are made.**

*- Page 12, line 5: Add "dust" in front of "sources".*
Done

*- Page 12, line 6: Put brackets around "5 a".*
Done

*- Fig. 5: Does it make sense to use the surface concentration for this plot? Wouldn't it be more reasonable to do the simulations at a higher altitude?*
We use the surface concentration as it is where most INP observation are made. Figure 6 show the vertical profiles of INP ambient.

We have addressed this in the caption of Fig 5 (see above).

*- Fig. 5a: What is the white spot in the plot (bottomleft)?*
It was a concentration range that was outside the colorbar range. Now it is corrected.

*- Fig 6: Add a label to the colorscale. Which variable is plotted?*
The description of the variable plotted ($[INP]_{ambient}$) is defined in the caption of the figure. We have added it to the colorscale label

*- Page 14, line 1, 4 and 5 and caption Fig. 6: You plot seasons and not separate months- adapt the wording.*
We have changed the 'monthly' to **'seasonal'**

*- Page 14, line 4: Add a space between "Fig." and "6".*
Done

*- Page 14, line 18: More consistent with what?*
Changed to prevalent

*- Page 14, line 22: Add brackets around "Fig. 6".*

Done

*- Fig. 6: It would be more consistent with the following analysis if you would give the INP conc. in 1/l instead of 1/m$_3$.*

We prefer to keep the units in 1/m3 in figure 6, as using 1/l would make the numbers too small affecting the quality of the image as the plot becomes too messy.

*- Fig. 6: Instead of the black contour lines you could also display two plots next to each other, that is maybe better readable. In the second plot the labels of the contour lines are difficult to read (overlap).*
We have modified the plots to avoid overlap between the labels of the contour lines, but would prefer to maintain one plot since it makes the comparison more direct.

*- Fig. 7: Why do you have values in the temperature range below -26◦C?*
The concentrations at temperatures colder than the limit of the parameterizations are set as the value at the limiting temperature as explained in Page 11,line 4:

*"The concentrations of [INP]ambient at temperatures colder than the temperature limit of the parameterizations (for K-feldspar: -25oC and marine organics: -27oC) is set at the value defined by the concentration at the limiting temperature of each parameterization. This is consistent with studies that caution against extrapolating singular parameterizations outside the range where measurements were made."*

*- Fig. 7: Especially in the third plot there are INP values even below -40◦C- you should explain these "artefacts" or whatever it is.*
The black lines in Figure 7 represent seasonal mean isotherms. Some of the values are below those lines because of day-to-day temperature variability.
We have rephrased the caption to clarify that they are seasonal mean isotherms.

*- Caption Fig. 7: Add a space between label "ambient" and "concentration" (line 2).*
Done

*- Page 17, line 1: Other schemes indirectly capture the source since large particles sediment and are more predominant close to the source region. Why is only a species-differentiating scheme able to capture variations and long-term trends?*
Because variations in aerosol emissions could be different for different aerosol species. This will imply that particles with very different ice nucleating abilities will be emitted in different amounts and hence the change in the INP concentrations will not necessarily be proportional to the change in the total emitted aerosol amount.  We have rephrased this sentence to clarify the concept.

**…so they may not capture variations and long-term trends since different aerosol types have different ice nucleating abilities.**

*- Page 17, line 6: Add a space between "Table" and "1" (remove the . or write Tab.). Add a space between "Fig." and "8c".*
Done

*- Page 17, line 10: There is no improvement shown in Tab. 1 (the unscaled values or not shown)? Eather add it in the table, or remove the reference to the table.*
The reference refers to the value of the correlation coefficient.

*- Page 17, line 23: Add brackets around "Tab. 1".*
Done

*- Table 1: Why is the correlation coefficient calculated for the logarithm of the values? Please explain shortly in the manuscript.*
It is calculated with the logarithm of the values as they vary logarithmically with temperature. Explanation added to caption.

**The correlation coefficient has been calculated with the logarithm of the values as INP concentrations vary logarithmically with temperature**

*- Page 18, line 10: Since you do not know if preferential INP in-cloud removal is important you should change "are" to "could be". Same in line 11 for the terrestrial source of INP.*
Done

There was a mistake in the text. In page 17 line 24, where it says Figure 8g it should say Figure 8f. There is where the figure was mentioned. It is been solved now that figure 8 has been divided in 2.

We have increased the font of the figures and divided figure 8 into 2 different figure so it improves its aspect.

Checked

Added [INP]

Done

*Done*

Done

*Expanded:*
**In addition, more measurements in the ambient atmosphere for different environments and seasons are necessary to better evaluate and constrain models. Among those, exploratory studies about the composition and type of ice nucleating particles in terrestrial environments at high temperatures will be crucial to determine which species need to be included in models.**

The derivation of the equation has been added

See next comment.

We have modified the caption to include the required information:
**a) OMF compared as a function of chlorophyll-a content and surface wind speed for the monthly mean values across the year in both stations. The size of the points represent the mean chlorophyll-a content of the grid-boxes related previously to every station (Fig.10), the colour of the points is related to the wind speed of those grid-boxes. b) Shows the performance of the parameterization for reproducing the OMF calculated with the simulated concentration of sub-micron sea-salt and the observed values of WIOM.**

*caption.*
Done
*- Appendix B: How do you get from Eq. B2 to B3?*
By adding  $f(0,\lambda)- f(0,\lambda)$ (equals 0) into the right side of the equation.  The first $f(0,\lambda)$ goes inside the sumatory (note the change in the stating value of the sumatory).

*- Page 26, line 1: Do you refer to size distribution when you write "distribution"?*
Yes, added 'size'

*- Appendix B: What does the last section mean in your model context?*
It means that for aerosol species with small number of active sites per particles ($\lambda<0.1$ always) we can reduce the complexity of the calculation. This method is used for marine organic aerosols as stated in Page 9 line 4

*- Table 2: Remove brackets around the references.*
Done

*- Table 2: Are the references unpublished where you did add the label "BACCHUS"? Otherwise I do not understand why this is labeled like this and what it means.*
The dataset was obtained from the BACCHUS project. The values are published, but the data was taken straight from the dataset.

We added a clarification:
**The datasets obtained through the BACCHUS project database are labelled as "BACCHUS" in table 2.**

*General remarks:*
*- The citations are not done consistent- sometimes brackets are used where there should not be, sometimes brackets are missing, e.g. at page 3 line 15 brackets are missing vs. at page 2 line 34 brackets should be removed. Please thoroughly go through the citations again.*
We have gone through that again. This was an issue with transferring text between latex and word.

*- Please add a space between numbers and units, e.g. page 5 line 7: 10 hPa.*
Done

*- Units should not be italic, e.g. page 6 line 5.*
Corrected

*- The naming of the modell is not consistent throughout the paper, sometimes you write GLOMAP, sometimes GLOMAP-mode. This should be explained (if the names are different on purpose) or made consistent.*
Checked.

*- Reduce the space between the single letters within your variables INP and ff, that increases the readability.*
Done
*- Be consistent with writing OMF as a variable in italic or not, e.g. page 24, line 3.*
Done

---

## Author Response (AR2)

We thank the review for their helpful and constructive comments.
The response is structured as in the previous response:

Blue for the reviewer comment

Normal text for our answers

**Bold for the changes in the manuscript**

- You stated in your abstract that "Many modelling studies use parameterizations for heterogeneous ice nucleation and cloud ice
processes that do not account for this difference because they were developed based on INP measurements made predominantly in terrestrial environments." You reacted on the argument that "If a scheme was developed for terrestrial sources and does not account for marine sources, the INP concentration represented in the model would be different for marine and terrestrial sources as well (it would be smaller above marine regions as shown by the field observations)." by stating that the parameterization schemes you referred to are aerosol-species-independend. In that case your argumentation is true. However, in the abstract (and also partly in the introduction) you write it so generally that it does not become clear like that. You either have to add the specific parameterization schemes that you refer to or you have at least to state that most parameterization schemes do not take the aerosol species (and thus indirectly the sources) into account. There are existing parameterization schemes used that take the aerosol composition into account and also only based on terrestrial sources show lower INP over the ocean.

We have rephrased the abstract to clarify what types of parameterizations we refer to as suggested by the reviewer adding a sentence to mention the fact that the aerosol composition is missing in many of the parameterizations.

**Many modelling studies use parameterizations for heterogeneous
ice nucleation and cloud ice processes that do not account for this
difference because they were developed based on INP measurements made
predominantly in terrestrial environments without considering the aerosol composition.**

- I think it is missleading how you formulated your statement that the particle-to-particle variability is better represented in the ns-scheme. It is true that the ns-approach takes into account the spectrum of a species in activating ice, but still in the end you use an average density of active sites. At one specific temperature you will get a fixed fraction of particles that nucleate- this number is the average number of particles in your total particle population that freeze under the given temperature. If you would want to explicitely account for (a variable) particle to particle variation you would need to use a ns-distribution for each temperature for each species.

By particle-to-particle variability we mean that just a fraction of particles is considered to nucleate ice in the singular description, whereas in classical nucleation theory long time integration will eventually allow all particles to nucleate ice.

We agree that different ns values for different species are necessary to account for the differences in their ice nucleating ability.

We have added some sentences to clarify what we mean by particle-to-particle variability and mention that species specific active sites densities are necessary to account for their different ice nucleating abilities:

**….particle-to-particle variability is not represented  (Herbert et al.2014)**
**as long time integration will eventually allow all particles to nucleate ice….**

**…is assumed to be of secondary importance**
**compared to the particle-to-particle variability (just a fraction of particles nucleate ice)…**

**…fraction of the particles will nucleate ice at a certain temperature (See Appendix:2).**
**In the case of having different aerosol species, a different density of active sites for every species has to be defined in order to account for their different ability to nucleate ice.**

- It is not true that in the case of using classical nucleation theory with a single contact angle time dependence is the main factor that drives nucleation, also in this case it is temperature. Time is less important compared to temperature (also in CNT).

We agree in this point. To clarify the idea we added "(although temperature is the main driver of nucleation)"

**This approach has the advantage that the time dependence of ice nucleation is represented (although temperature is the main driver of nucleation), but when a single contact angle is used to describe ice nucleation by a single aerosol species…**

- Page 3, line 24: Change "The temperature dependent number of active sites..." to "The temperature dependent number of active sites per surface area ..."

Added: **per surface area**

- Table 1: You could use the three different results coming from different temperature ranges (without *, *, **) to analyse how the different parameterizations react on extrapolation. The Niemand et al. scheme seems to have the trouble predicting the INP at low temperatures, the perfomance improves in the case the cold temperature values are left away. The marine and k-feldspar scheme on the other hand seems to especially capture the high temperature range well, since the performance goes down in the case of leaving the high temperature values.

We have included in the discussion of the performance of Niemand's parameterization the following sentence:

**A similar trend is observed when this parameterization**
**is extrapolated to higher temperatures.**

We prefer to limit the discussion on the effect of extrapolating the parameterization to higher temperatures to just a few sentences, as the number of datapoints at those temperatures is quite small, so it is hard to conclude that this behaviour would be maintained in the future if more observations are included.

- Looking at Fig. 8: It could have been interesting to see the same for a k-feldspar only parameterization scheme (without marine contribution).

An estimate of the contribution of k-feldspar particles to the simulated concentration in every datapoint is shown in figure 9 b

- Page 10, line 23: "through all the mixed-phase range of temperatures" sounds a bit strange. You mean "through all the temperature range of mixed-phase clouds"?
Corrected

- Fig. 3: The example of color scale for the temperatures in mixed-phase cloud still does not make sense or does not add any value to the Figure. You either have to use the colors somewhere in the Figure, or leave them away.
We have deleted the colorscale of the figure

- Caption Fig. 3: ambient should be subscript.
Done

- Caption Fig. 4: 600hpa should be 600 hPa.
Done

- In Fig. 4 you write that the upper panel shows the INP conc. at 600 hPa at -20°C and the lower panel the INP conc. at 600 hPa at ambient temperature. If one looks at the Arctic, the INP conc. is higher in the lower panel compared to the upper panel. That means the temperature has to be lower in the lower panel and thus in the Arctic the local ambient temperature is lower than -20°C? Is that the case or is something in the caption/figure wrong?
Yes, the temperature is most of the time lower than -20$^o$C in the Artic at 600hpa. You can see the modelled isotherms in figure 7.

- I understand why you chose a temperature of -15°C and the surface concentrations to calculate INP ambient and compare it to the measurements. However, that should be critically commented in the paper since the real (ambient) IN conc. would be different from what you call INP ambient.

A discussion on the 2 different ways of looking at INP concentrations is included in page 10 lines 12 to 28. Also figure 3 explains the difference between what we call [INP]_ambient and [INP]_T.

For calculating [INP]_ambient, we use the local temperature, and for [INP]_T we use a defined temperature.
Figure 5 shows [INP]_T at a temperature of -15 $^{o}$C: [INP]_-15 as stated in the colorbar label and in the figure caption.

- Figure 6: You could rename the axis label 1/m³ to 10^(-3) 1/l. Then you would keep it consistent and the numbers would still be readable in the plot.

After trying several ways of plotting it, the readability did not improve. When the units are converted to liters instead of m-3, the decimal values and the floating point together with the contourlines make the plot very messy as they mix too much.
 However, we have decrease slightly the size of the numbers in the contourlines, so now they do not mix and the plot look a bit less messy.

- Page 12, line 5/6: Provide a citation for studies about a statement about extrapolating singular parameterizations.
The citation is included in the following sentence.

"This is consistent with studies that caution against extrapolating singular parameterizations outside the range where measurements were made. For example, Niedermeier et al. (2015)…"

- Page 23, line 11: Add INP -> "In addition, more INP measurements...".
Done

[revised manuscript text omitted]